# Missing OH Reactivity in the Global Marine Boundary Layer

Alexander B. Thames[1], William H. Brune[1], David O. Miller[1], Hannah M. Allen[2], Eric C. Apel[3], Donald R. Blake[4], T. Paul Bui[5], Roisin Commane[6], John D. Crounse[7], Bruce C. Daube[8], Glenn S. Diskin[9], Joshua P. DiGangi[9], James. W. Elkins[10], Samuel R. Hall[3], Thomas F. Hanisco[11], Reem A. Hannun[11;12], Eric Hintsa[10;13], Rebecca S. Hornbrook[3], Michelle J. Kim[7], Kathryn McKain[10;13], Fred L. Moore[10;13], Julie M. Nicely[11;14], Jeffrey Peischl[10;15], Thomas B. Ryerson[15], Jason M. St. Clair[11;12], Colm Sweeney[10], Alex Teng[2], Chelsea R. Thompson[13;15], Kirk Ullmann[3], Paul O. Wennberg[7,16], and Glenn M. Wolfe[11;12]

[1]Department of Meteorology and Atmospheric Science, The Pennsylvania State University, University Park, PA, USA.
[2]Division of Chemistry and Chemical Engineering, California Institute of Technology, Pasadena, CA, USA.
[3]Atmospheric Chemistry Observations & Modeling Laboratory, National Center for Atmospheric Research, Boulder, CO, USA.
[4]Department of Chemistry, University of California, Irvine, CA, USA.
[5]Earth Science Division, NASA Ames Research Center, Moffett Field, CA, USA.
[6]Department of Earth and Environmental Sciences, Lamont-Doherty Earth Observatory, Columbia University, Palisades, NY, USA.
[7]Division of Geological and Planetary Sciences, California Institute of Technology, Pasadena, CA, USA.
[8]Department of Earth and Planetary Sciences, Harvard University, Cambridge, MA, USA.
[9]Chemistry and Dynamics Branch, NASA Langley Research Center, Hampton, VA, USA.
[10]Global Monitoring Division, NOAA Earth System Research Laboratory, Boulder, CO, USA.
[11]Atmospheric Chemistry and Dynamics Laboratory, NASA Goddard Space Flight Center, Greenbelt, MD, USA.
[12]Joint Center for Earth Systems Technology, University of Maryland, Baltimore County, Catonsville, MD, USA.
[13]Cooperative Institute for Research in Environmental Sciences, University of Colorado, Boulder, CO, USA.
[14]Earth System Science Interdisciplinary Center, University of Maryland, College Park, MD, USA
[15]Chemical Sciences Division, NOAA Earth System Research Laboratory, Boulder, CO, USA.
[16]Division of Engineering and Applied Science, California Institute of Technology, Pasadena, CA, USA.

*Correspondence to*: William H. Brune (whb2@psu.edu)

**Abstract.** The hydroxyl radical (OH) reacts with thousands of chemical species in the atmosphere, initiating their removal and the chemical reaction sequences that produce ozone, secondary aerosols, and gas-phase acids. OH reactivity, which is the inverse of OH lifetime, influences the OH abundance and the ability of OH to cleanse the atmosphere. The NASA Atmospheric Tomography (ATom) campaign used instruments on the NASA DC-8 aircraft to measure OH reactivity and more than 100 trace chemical species. ATom presented a unique opportunity to test the completeness of the OH reactivity calculated from the chemical species measurements by comparing it to the measured OH reactivity over two oceans across four seasons. Although, throughout much of the free troposphere, the calculated OH reactivity was below the limit-of-detection for the ATom instrument used to measure OH reactivity, the instrument was able to measure the OH reactivity in and just above the marine boundary layer. The mean measured value of OH reactivity in the marine boundary layer across all latitudes and all ATom deployments was 1.9 s$^{-1}$, which is 0.5 s$^{-1}$ larger than the mean calculated OH reactivity. The missing OH reactivity, the difference between the measured and calculated OH reactivity, varied between 0 s$^{-1}$ to 3.5 s$^{-1}$, with the highest values over the Northern Hemisphere Pacific Ocean. Correlations of missing OH reactivity with formaldehyde, dimethyl sulfide, butanal, and sea surface temperature suggest the presence of unmeasured or unknown volatile organic compounds or oxygenated volatile organic compounds associated with ocean emissions.

## 1 Introduction

The primary fate of the thousands of trace gases emitted into the atmosphere is chemical reaction with the hydroxyl radical (OH). While OH is produced primarily by the photolysis of ozone, followed by a reaction between excited-state atomic oxygen and water vapor, OH is lost at the rate determined by the sum of the reaction frequencies with these trace gases. This sum of loss frequencies is called the OH reactivity and has units of s[-1]. If OH production remains constant, increases in OH reactivity will decrease the total atmospheric OH concentration. Thus, understanding global OH reactivity is a key to
understanding global OH and the global atmospheric oxidation capacity.

An important example is methane ($CH_4$), which is removed from the atmosphere primarily by reaction with OH. Two estimates of the $CH_4$ lifetime due to oxidation by OH are 9.7 ± 1.5 years (Naik et al., 2013) and 11.2 ± 1.3 years (Prather et al., 2012). A recent global inverse analysis of GOSAT satellite $CH_4$ column emissions finds a $CH_4$ lifetime of 10.8 ± 0.4
years for oxidation by tropospheric OH (Maasakkers et al., 2019), which is within the uncertainties of the other two estimates. Understanding the $CH_4$ lifetime depends on understanding global spatial and temporal OH distributions, which are strongly influenced by the spatial and temporal distribution of OH reactivity.

OH reactivity is the inverse of the OH lifetime. It is calculated as a sum of OH reactant concentrations multiplied by their
reaction rate coefficients:

$$k_{OH} = \sum_i k_{(OH+X_i)}[X_i]. \tag{1}$$

where $k_{(OH+X_i)}$ represents some species X's reaction rate coefficient with OH and $[X_i]$ is the concentration of that species. If there is no OH production, then the equation for the OH decay is


$$\frac{d[OH]}{dt} = -k_{OH}[OH]. \tag{2}$$

The first direct measurements of OH reactivity were made in Nashville, TN in summer 1999 (Kovacs et al., 2003). The measured OH reactivity exceeded the calculated by about 30%, which was thought to come from short-lived highly reactive volatile organic compounds (VOCs) that were not measured in that study. The difference between the measured and
calculated OH reactivity was referred to as the "missing" OH reactivity. For forest environments, the first evidence for missing OH reactivity came from direct OH reactivity measurements in a northern Michigan forest in summer 2000 (Di Carlo et al., 2004). As much as a third of the OH reactivity was missing, with missing OH reactivity increasing with temperature in a manner identical to the expected increase of forest monoterpene emissions. Since then, OH reactivity has been measured many times in various urban, rural, and forest environments (Yang et al., 2016, and references therein). The
fraction of missing OH reactivity in different forests varies from less than 20%, which is approximately the uncertainty in the measured and calculated OH reactivity (Kaiser et al., 2016; Zannoni et al., 2016), to more than 50% (Nölscher et al., 2012; Nölscher et al., 2016). Considering the large numbers of trace gases emitted into the atmosphere (Goldstein and Galbally, 2007), it is possible that missing OH reactivity comes from OH reactants that were not measured or not included in previously calculated totals of the OH reactivity sum. In some studies, the OH reactants have been only those that were
measured, and in other studies unmeasured but modeled OH reactants – such as organic peroxyl radicals and oxygenated volatile organic compound (OVOC) products – have been included. A recent intercomparison of several OH reactivity instruments demonstrated that these large missing OH reactivity values are probably not due to instrument issues (Fuchs et al., 2017). These discrepancies have yet to be resolved.

One regime that has yet to be adequately investigated is the remote marine boundary layer (MBL) and the free troposphere above it, which comprises 70% of the global lower troposphere. Two prior studies measured OH reactivity in the MBL. The most recent was shipborne across the Mediterranean Sea, through the Suez Canal, and into the Arabian Gulf in summer 2017 (Pfannerstill et al., 2019). Several portions of this journey were heavily influenced by petrochemical activity or ship traffic, while others were relatively clean. Median measured OH reactivity for the different waterways ranged from 6 s[-1] to 13 s[-1],
while median calculated OH reactivity ranged from 2 s[-1] to 9 s[-1]. When more than 100 measured chemical species were

included in the calculated OH reactivity, the difference between the measured and calculated OH reactivity was reduced to being with measurement and calculation uncertainty for some regions, but significant missing OH reactivity remained for other regions. In the cleaner portions of the Mediterranean and Adriatic Seas, the calculated OH reactivity of ~2 $s^{-1}$ was below the instrument's limit of detection (LOD = 5.4 $s^{-1}$).


The other study involved airborne OH reactivity measurements made during the Intercontinental Chemical Transport Experiment Phase B (INTEX-B) study, a NASA airborne campaign investigating Asian-influenced pollution over the north Pacific Ocean in April-May, 2006 (Mao et al., 2009). In this study, aged pollution plumes from Southeast Asia were encountered within large regions of relatively clean air. At altitudes below ~2 km, missing OH reactivity was ~2.4 $s^{-1}$, more

than the calculated OH reactivity of $1.6 \pm 0.4$ $s^{-1}$. It decreased to within measurement uncertainty above 4 km. The correlation of missing OH reactivity with formaldehyde (HCHO) suggested that the missing OH reactivity was due to highly reactive VOCs that had HCHO as a reaction product. The confinement of the missing OH reactivity to the MBL and just above it suggested that the cause of the missing OH reactivity was ocean emissions of volatile organic compounds (VOCs).

In this paper, we describe measurements of OH reactivity that were made during the NASA Atmospheric Tomography (ATom) campaign (ATom, 2016). This campaign took place in four month-long phases, each in a different season, between August 2016 and May 2018 and covered nearly all latitudes over the Pacific and Atlantic Oceans. Although the calculated OH reactivity in the middle-to-upper troposphere is less than the OH reactivity instrument's LOD of ~0.4 $s^{-1}$ at 68% confidence, this instrument can measure OH reactivity in and just above the MBL. The comprehensive instrument suite

deployed aboard the NASA DC-8 airborne laboratory allows a detailed examination of which trace gases most influence measured OH reactivity.

**2 Methods**

Here we discuss the ATom campaign, the OH reactivity instrument and its measurement capabilities, the model used to generate calculated OH reactivity, and the statistical analysis that was used to find correlations with missing OH reactivity.

**2.1 ATom**

The ATom campaign consisted of four deployments over all four seasons, starting with Northern Hemisphere summer in 2016 and ending with Northern Hemisphere spring in 2018 (Table 1).

Each deployment used the NASA DC-8 Airborne Science Laboratory (DC-8) to profile the atmosphere by frequently

ascending and descending between 0.2 km and 12 km on flights north from California to Alaska, down the Pacific to New Zealand, across the Antarctic Circle to Chile, up the Atlantic Ocean to Greenland, across the Arctic Circle to Alaska and then back to California (yellow lines in Fig. 1). As shown in Table 2, the DC-8 carried a suite of instruments that measured over 100 different chemical constituents, aerosol particle properties and chemical composition, photolysis frequencies, and meteorological variables (Wofsy et al., 2018; ATom, 2016).

**2.2 OH Reactivity Measurement**

The OH reactivity concept and the basic instrument have been described before for ground-based operation (Kovacs and Brune, 2001) and for aircraft operation (Mao et al., 2009). The instrument used for ATom, called OH Reactivity (OHR), is a version of the one described by Mao et al. (2009). A brief description of the concept and the instrument is presented below.

Sampled air is brought into the instrument during flight by ram force at the 1.2 cm diameter inlet and the Venturi effect at the instrument outlet. A movable wand at the center of a flow tube (7.5 cm dia.) injects OH into the flow tube at different distances from an OH detection inlet and axis similar to the one used to detect OH in the atmosphere. In the wand, OH is generated in a flow of humidified carrier ($N_2$ or purified air), which is exposed to 185 nm radiation from a Hg lamp that

photolyzes the $H_2O$ to make OH and $HO_2$. As the wand moves away from the detection axis, the signal observed of unreacted OH with the sample air decreases. Assuming a constant decay rate, measured OH reactivity is determined by Eq. (3):

$$k_{OH} = \frac{ln\left(\frac{[OH_o]}{[OH]}\right)}{\Delta t} - k_{offset} \tag{3}$$

where [OH] is the instantaneous OH concentration, $[OH_0]$ is the initial OH concentration, $\Delta t$ is reaction time between the [OH] measurements (the distance the wand moves divided by the flow speed), and $k_{offset}$ is the instrument offset due to OH loss to the walls or to impurities in the carrier gas. The wand moves approximately 10 cm in total along its path from closest point to farthest point from the detection axis. The sampling time step is synced with the Airborne Tropospheric Hydrogen Oxides Sensor (ATHOS, an instrument used in tandem with the OHR instrument to measure in situ OH and $HO_2$), which samples at 5 Hz. Depending on the ATom deployment, the wand takes 15 or 20 seconds to move 10 cm and back to its 145 starting position, where it rests for 5 or 10 seconds while the OH detection system switches the laser wavelength to off resonance with the OH absorption line to measure the signal background. Flow speeds through the OHR instrument are measured with a hot-wire anemometer and are typically between 0.25 m s$^{-1}$ at lower altitudes and 0.45 m s$^{-1}$ at higher altitudes, resulting in a typical total measured reaction time between 0.40 and 0.22 seconds.

It is important to note that all OH reactivity instruments measure the "instantaneous" OH reactivity, which is only the reactions that occur within the maximum reaction time observed by that instrument. This maximum time is typically less than a second. These instruments do not measure either subsequent OH reactivity or OH production if the time constants for that chemistry are greater than the maximum reaction time. In relatively clean environments, no subsequent chemistry affects the measured OH decay. However, in environments where NO is greater than a few ppbv, the reaction $HO_2 + NO \rightarrow OH +$ 155 $NO_2$ is fast enough to convert $HO_2$ to OH, thereby altering the observed OH decay. No high NO environments were encountered in ATom.

In all previous ground-based and aircraft-based studies, high purity $N_2$ was used as the carrier gas in the wand. During aircraft-based studies, a cylinder of $N_2$ gas was consumed on each 8-hour flight and accordingly had to be replaced before 160 the next flight. It was not possible to position caches of $N_2$ cylinders at each of the ~12 layovers during each ATom phase. Instead of $N_2$, air from a zero-air generator (PermaPure ZA-750-12) was used as the carrier gas in the laboratory prior to ATom. Before each mission, the zero-air generator media (Perma Pure ZA-Catalyst – Palladium on Aluminum Oxide) was replaced and the air produced by the unit was verified to be pure by running it through a Potential Aerosol Mass chamber that rapidly oxidizes any VOCs to particles (Lambe et al., 2011). No particles were seen, indicating that the air had 165 negligible amounts of larger reactive VOCs. The results of this test were consistent with those obtained by substituting the air from the zero-air generator with high purity nitrogen. The exception to this procedure was during ATom4, when the zero-air generator itself had to be replaced late in instrument integration period. The media was changed prior to the ATom4 integration and the research flights, but the air purity was unable to be checked until after the ATom4 deployment had ended, when it was found that the OHR offset was higher than in previous ATom deployments.

**2.3 OH Reactivity Measurement Offset Calibrations**

The OHR offset varied between the 4 ATom deployments due to changes in the zero-air generator performance and between research flights due to internal contamination from pre-flight conditions. These changes were tracked with measurements of the OHR instrument offset in the laboratory and, for ATom4, in situ during several flights. For the laboratory calibrations, the internal pressure of the OHR instrument was varied between 30 and 100 kPa to simulate in-flight conditions. For the in 175 situ calibrations taken during the second half of ATom4, the OHR instrument was switched from sampling the ambient flow to sampling high purity $N_2$ from a reserve $N_2$ cylinder. The flow rate out of the cylinder was adjusted to match the flow speed measured by the hot-wire anemometer just prior to the switch. During ascent and descent, the internal pressure and flow speed changed too quickly over the length of one decay to get good offset calibrations, so offset calibrations were taken only from stable altitude legs, predominantly at the low altitudes.


Two complete laboratory calibrations, the in situ calibration, and a calibration only at ~100 kPa were used to determine $k_{offset}$ for the different ATom deployments (Fig. 2). The 2017 calibration applies to ATom2 and ATom3, while the 2018 calibration applies to ATom4. For ATom1, the offset was calibrated at only 97 kPa prior to the mission, but it is in excellent agreement with the offset calibrated for ATom4. We can safely assume that the ATom4 offset slope can be applied to ATom1 because all offset calibrations performed at low OHR flow tube pressures, even those of Mao et al. (2009), give ~2 s[-1] for the offset value. The difficulty of maintaining steady calibration conditions in flight during ATom4 caused the large in situ calibration error. The standard deviation of these in situ offset calibrations is 0.75 s[-1], which is 2.5 to 3 times larger than the standard deviation obtained for ambient measurements in clean air for the same altitude and number of measurements, indicating that the atmospheric measurement precision is much better than could be achieved in these difficult offset calibrations. Yet even with this lower precision, the mean in situ offset at high and low pressure agree with the linear fit of the laboratory calibrations to within 20% at low pressures and 3% at high pressure. The excellent agreement between the laboratory and in situ offset calibrations for ATom4 confirms the finding of Mao et al. (2009) that laboratory offset calibrations before or after a campaign accurately capture the instrument offset during the campaign.

This observed pressure dependence of the offset calibration is different from the behavior of the pressure-independent offset calibration used by Mao et al. (2009). However, a re-examination of the Mao et al. (2009) data indicates that the offset during INTEX-B was also pressure dependent, with an offset of 2.9 s[-1] at high OHR flow tube pressure and 2.0 s[-1] at low OHR flow tube pressure, nearly identical to the values used for ATom2/ATom3.

The difference in the linear fit to the offset calibration for ATom1 and ATom 4 and the linear fit to the offset calibration for ATom2 and ATom3 is pressure dependent (Fig. 2). The standard volume airflow in the wand was constant, but the ambient volume flow in the flow tube decreased by a factor of ~2 as the flow tube pressure increased from 30 kPa to 100 kPa. As a result, the contamination concentration from the wand air also increased a factor of ~2 as flow tube pressure increased. This pressure-dependent contamination concentration explains much of the difference between the two fitted lines and provides evidence that contamination in the wand flow was a substantial contributor to the changes in the zero offset between ATom1/ATom4 and ATom2/ATom3. The good agreement between the fit for ATom2/ATom3 and the offset calibrations of Mao et al. (2009), who used ultra-high purity $N_2$, suggests that the zero air for ATom2/ATom3 had negligible contamination.

The OHR instrument zero offset varied slightly from flight to flight because the variable air quality produced by the zero-air generator. This flight-to-flight variation was tracked and the OH reactivity offset was corrected by the following procedure. The OH reactivity calculated from the model at the OHR instrument's temperature and pressure (see Section 2.5) was 0.25-0.30 s[-1] for the upper troposphere during all ATom deployments and latitudes. The offset calibrations were adjusted in the range of 0.34±0.32 s[-1] for each research flight by a pressure-invariant offset that was necessary to equate the mean measured and model-calculated OH reactivity values for data taken above 8 km altitude. If this offset correction is not used for all altitudes, then the OH reactivity in the 2-8 km range varies unreasonably from flight-to-flight, even going significantly negative at times. In effect, we used the upper troposphere as a clean standard in order to fine-tune $k_{offset}$, just as Mao et al. (2009) did.

The OH signals in the upper troposphere were high enough to allow the slopes of the OH decays to be determined with good precision for each 20-30 s decay. However, at the low altitudes, quenching of the fluorescence signal by higher air and water vapor abundances reduced the OH signals. To compensate for this reduction, decays were binned into 1-minute sums before the decay slopes were calculated. Three OH signal decays from low altitudes during ATom2 are shown in Fig. 3. When $k_{offset}$ is subtracted from the decays shown in Fig. 3, their values become ~5 s[-1] (blue), ~3 s[-1] (teal), and ~2 s[-1] (yellow).

## 2.4 Missing OH Reactivity Uncertainty Analysis

The uncertainty for missing OH reactivity in the MBL at the 68% confidence level comes from four components: the decay measurement itself; the offset as determined by the slope and intercepts of the fits to the laboratory OH reactivity offset calibrations (Fig. 2); the flight-to-flight offset variation as judged by fitting the measured OH reactivity to the model-calculated OH reactivity at 8-12 km altitude; and the model calculations. First, the uncertainty in decay fit is approximately

±7.5%, which for a typical OH reactivity measurement in the MBL of ~2 s$^{-1}$, would give an uncertainty of ±0.15 s$^{-1}$.
Second, the uncertainty in the OH reactivity offset in the MBL is found from the sum of the slope uncertainty times the OHR flow tube pressure, which is ~100 kPa in the MBL, (±0.16 s$^{-1}$) and the intercept uncertainty (±0.11 s$^{-1}$). The two uncertainties are assumed to be correlated. Third, the uncertainty in the flight-to-flight offset variation is the standard deviation of the mean for each high altitude short level leg (±0.15 s$^{-1}$). Fourth, the uncertainty of the model-calculated OH reactivity was determined by Eq. 4:

$$\Delta k_{OH}^{calc} \ (s^{-1}) = \sqrt{\sum [(k_i \Delta x_i)^2 + (\Delta k_i x_i)^2]} \qquad (4)$$

where $k_i$ are the reaction rate coefficients and $x_i$ are the OH reactant concentrations. The rate coefficient uncertainties come from Burkholder et al. (2016) and the chemical species uncertainties come from Table 2 and Brune et al. (2020). For the 11 chemical species responsible for 95% of the total OH reactivity in the MBL, this uncertainty is ±0.08 s$^{-1}$. The square root of the sum of the squares of all these uncertainties yields a total uncertainty for the MBL missing OH reactivity of ±0.32 s$^{-1}$ at the 68% confidence level.

The OH reactivity from the model at the ambient temperature and pressure rarely exceeded 2 s$^{-1}$ in the planetary boundary and was only 0.2 s$^{-1}$ in the free troposphere. These low values presented a significant challenge for our OHR instrument, as it would have for any OH reactivity instrument; even the most precise instrument measuring in a chamber at its home laboratory has a LOD of ±0.2 s$^{-1}$, 68% confidence, for a measurement integration time of 60-160 seconds (Fuchs et al., 2017). If the same instrument were to sample air masses on an aircraft traveling 200 m s$^{-1}$, its precision would likely be degraded. From this perspective, the LOD on these ATom measurements is about as low as that for any other OH reactivity measurements.

The analysis in the paper is focused on the first three ATom deployments. ATom4 is excluded from this analysis for two reasons. First, offset calibrations were performed during more than half of the low-altitude periods over the Atlantic, severely limiting the ambient measurements in the MBL. Second, the increased contamination during ATom4 made the OH reactivity measurements 2-3 times noisier than during the previous ATom deployments.

**2.5 Photochemical Box Model**

The photochemical box model used to calculate OH reactivity is the Framework for 0-D Atmospheric Modeling (F0AM) (Wolfe et al, 2016). It uses the Master Chemical Mechanism v3.3.1 (MCMv331) for all gas-phase reactions (Jenkin et al., 1997; Saunders et al., 2003). Both the F0AM model framework and MCMv3.3.1 are publicly available. The reactions of $CH_3O_2 + OH$ and $C_2H_5O_2 + OH$ were added to the model mechanism with rate coefficients from Assaf et al. (2017). The model was run with the integration time set to 3 days with a first-order dilution lifetime of 12 hours, although the calculated OH reactivity was the same to within a few percent for an order-of-magnitude change in these times. The model was constrained by the simultaneous measurements listed in Table 2. These measurements were taken from the 1-second merge file, averaged to 1-minute values, and interpolated to a common 1-minute time step. In cases where multiple measurements of a chemical species exist (e.g., CO), a primary measurement was chosen and other measurements were used to fill gaps in the primary measurement.

To compare measured and calculated OH reactivity, the model-calculated OH reactivity must be corrected to the OHR flow tube pressure and temperature. For the rest of this paper, "calculated OH reactivity" will refer to these corrected values. Equation 1 was then used to find the calculated OH reactivity. If the measured and calculated OH reactivity agreed, then there was no missing OH reactivity to within the uncertainties for both the measured and the calculated values. However, if there was missing OH reactivity in the flow tube, then the missing OH reactivity in the atmosphere may be different because the temperature dependence of its reaction rate coefficients is unknown. Fortunately, the focus of this study is in and just above the MBL where the flow tube pressures and temperatures are nearly identical to atmospheric temperatures and pressures. The OH reactivity calculated from the model output at the flow tube pressure and temperature is within ±10% of

that calculated at ambient conditions. Thus, the missing OH reactivity at the flow tube temperature and pressure is assumed to be equal the atmospheric missing OH reactivity.

If missing OH reactivity is found, a likely source is unknown VOCs or OVOCs, which we will call X. The abundance of X
was determined from the missing OH reactivity by Eq. (5).

$$X = \frac{mOHR}{k_{X+OH}} \frac{10^9}{M}$$ (5)

where $X$ is the missing chemical species (ppbv), $mOHR$ is the missing OH reactivity, $k_{X+OH}$ is the reaction rate coefficient
for the reaction X + OH → products, and $M$ is the air number concentration. We assume that $k_{X+OH} = 10^{-10}$ cm$^3$ s$^{-1}$, which gives X a lifetime of about an hour for the typical daytime [OH] of ~$3 \times 10^6$ cm$^{-3}$. For these assumptions, an X abundance of 400 pptv corresponds to a missing OH reactivity of 1 s$^{-1}$. This arbitrary rate coefficient approximates a rate coefficient for a reaction of a sesquiterpene with OH. If X is an alkane or alkene that has a lower reaction rate coefficient, then the required X abundance would be larger.

Simple X oxidation chemistry was added to the photochemical mechanism to test the impact of X on the modeled OH and HO$_2$. This assumed additional chemistry is provided in Table 3. XO$_2$ is used to designate the peroxy radical formed from X. Rate coefficients for CH$_3$O$_2$ and CH$_3$OOH were assumed to apply to XO$_2$ and XOOH. Case 1 assumes that no OH is regenerated in the oxidation sequence for X, while case 2 assumes that OH is regenerated for every oxidation sequence of X.

**2.6 Correlation Analysis**

An analysis of correlations between missing OH reactivity and the chemical or environmental factors could indicate possible causes of the missing OH reactivity. Linear regressions were found for missing OH reactivity and every measured and calculated chemical species and meteorological parameter. Calculated chemical species with abundances less than 1 pptv were not included in the regressions. SST and chlorophyll data come from NASA Earth Observations (2019). Correlations
were performed on the first three ATom deployments individually and the first three ATom deployments combined.

To reduce the noise in the missing OH reactivity values prior to doing any correlation analysis, the 1-minute missing OH reactivity values were averaged into "per-dip" bins and "per-flight" bins. The term "per-dip" means that the missing OH reactivity was averaged over each dip, typically a 5-minute level-altitude leg at 160 m. The term "per-flight" means that the
missing OH reactivity for all the dips in a flight were averaged together. The standard deviation of the 1-minute measurements within each dip was typically 0.4 s$^{-1}$, while the standard deviation of the per-dip measurements in a flight was 0.25 s$^{-1}$. The low-level legs used for the per-dip means were generally in the MBL because its height was greater than 160 m 85% of the time. The MBL height is the altitude below which the potential temperature is constant. Each per-flight bin is the mean of each flight's per-dip set. A per-dip bin occurred roughly every hour of flight. Each per-flight bin spanned only a few
degrees of latitude near the poles but as much as 50° of latitude in the tropics. Only the measurements made while flying over the ocean were included in the per-dip and per-flight averaging because the dips over land sampled terrestrial or ice emissions and not ocean emissions.

Individual measured or calculated meteorological parameters and chemical species passed a preliminary correlation
threshold for missing OH reactivity if the sign of each regression was the same for ATom1, ATom2, and ATom3. Correlations that passed this preliminary filter had their R$^2$ values averaged between each deployment individually and grouped together. The averaged correlation coefficients were then tallied and ranked from greatest to least R$^2$. The top 10% of these correlations for both the per-dip and per-flight averages were combined into one data set. Because the missing OH reactivity showed some latitude dependence, the same multi-step technique was performed on all the chemical species and
meteorological parameters in different hemispheres: Northern, Southern, Eastern, and Western. Both data sets were then combined into a single data set and the strongest of these correlations were reported.

## 3 Results

The focus of these results is the OH reactivity measurements in and just above the MBL. However, the OH reactivity measurements are shown for the entire range of altitudes, even though the high-altitude (>8 km) OH reactivity values were set to the calculated OH reactivity that was corrected to the OHR flow tube pressure and temperature.

### 3.1 Global OH Reactivity Versus Altitude

The average calculated global OH reactivity at the lowest altitudes is about an order of magnitude less than the average OH reactivity in cities or forests (Yang et al., 2016), which is typically 10-50 $s^{-1}$. For ATom, calculated OH reactivity is less than 2 $s^{-1}$ averaged over all latitudes and seasons (Fig. 4). In different seasons and regions, this calculated OH reactivity consists of CO (30-40%), $CH_4$ (19-24%), methyl hydroperoxide (MHP) (5-16%), aldehydes (11-12%), $H_2$ (6-7%), $O_3$ (2-5%), and $HO_2$ (2-6%), $H_2O_2$ (0-5%), and $CH_3O_2$ (0-7%), with the remaining reactants totaling less than 10%. The ordering of these reactants is similar to that of Mao et al. (2009), although in their work the calculated OH reactivity due to CO was about 60%, $CH_4$ about 15%, and all OVOCs about 16%. Part of this difference can be ascribed to more OVOC measurements in ATom and the greater CO abundances in the Northern Hemisphere where INTEX-B occurred.

To compare measured and calculated OH reactivity, the calculated OH reactivity must be corrected to the flow tube pressure and temperature. For the rest of this paper, "calculated OH reactivity" will refer to these corrected values. The calculated OH reactivity decreases from ~1.5 $s^{-1}$ in the MBL to 0.25-0.30 $s^{-1}$ in the upper troposphere (Fig. 4). The mean measured OH reactivity has been matched to the mean calculated OH reactivity for altitudes above 8 km, but the two are independent at lower altitudes. The mean measured and calculated OH reactivity agree to within combined uncertainties for altitudes between 8 km and 2-4 km, but the mean measured OH reactivity becomes increasingly greater than the mean calculated below 2-4 km and especially in the MBL. However, the differences in the mean values is not the best way to understand these differences between measured and calculated OH reactivity.

### 3.2 Missing OH Reactivity: Statistical Evidence

A better approach is to find the missing OH reactivity for each measurement point and then compare the mean values. The missing OH reactivity is plotted as a function of altitude for ATom1, ATom2, and ATom3 (Fig. 5). The mean missing OH reactivity is set to 0 $s^{-1}$ for 8-12 km, remains near to 0 from 8 to 2-4 km, and increases below 2-4 km. The 1-minute measurements are a good indicator of the measurement precision, which is $\pm0.35$ $s^{-1}$ for ATom1 and $\pm0.25$ $s^{-1}$ for ATom2 and ATom3.

In the MBL, the mean missing OH reactivity is 0.4 $s^{-1}$ for ATom1, 0.5 $s^{-1}$ for ATom3, 0.7 $s^{-1}$ for ATom2. From a Student t-test in which the MBL missing OH reactivity is compared to either the values at 6-8 km or 8-12 km altitude ranges, the differences in mean missing OH reactivity between the MBL and the higher altitudes is statistically significant for a significance level, $\alpha$, equal to 0.01, with p-values $< 10^{-15}$. However, the mean MBL missing OH reactivity values are close to the upper limit on the absolute missing OH reactivity uncertainty (95% confidence), which is 0.64 $s^{-1}$ (blue bar, Fig. 5). There is a small probability (2-10%) that the mean MBL missing OH reactivity is due only to absolute error in the missing OH reactivity measurement that was derived in Section 2.4.

The mean MBL missing OH reactivity contains measurements for which the missing OH reactivity is 0 $s^{-1}$. The real interest is in the missing OH reactivity that is greater than can be explained by absolute missing OH reactivity measurement error or precision. From Fig 5., it is clear that the positive scatter of data is much greater than the negative. The means of standard deviations of the negative values and of the positive values were calculated for 1-km height intervals (dashed lines). These lines and the individual data points both indicate skewness in the missing OH reactivity, especially in the lowest 2-4 km altitude. A skewness test shows that, in and just above the MBL, missing OH reactivity from ATom1 and ATom2 exhibit weak-to-moderate skewness (~0.4) in the MBL while missing OH reactivity from ATom2 exhibits strong skewness (1.4).

Quantile-quantile plots (Q-Q plots) provide a visual description of the relationship between a sample distribution and a normal distribution. The standard normal quantiles are plotted on the x-axis and the sample quantiles on the y-axis. If the sample is perfectly normally distributed, then its values will lie along a straight line. Data lying higher than the line for values on the right side of the normal distribution (positive standard normal quantiles) indicate more high-value data than expected, while data higher than the line for values on the left side of the normal distribution (negative standard normal quantiles) indicate fewer low-value data than expected.

Q-Q plots are shown for three ATom2 cases in Fig. 6. The large boxes are the interquartile range between the $1^{st}$ quartile (25% of the data below it) to the $3^{rd}$ quartile (75% below). The missing OH reactivity data for altitudes greater than 8 km (red data) is normally distributed until the standard normal quantile of 2, meaning that less than a few percent of the data is higher than expected. On the other hand, the missing OH reactivity data in the MBL (blue data) are normally distributed between standard normal quantiles of -2 and 1, meaning that a few percent of low-value are less than expected, but, more importantly, as much as 20% of the high-value data is greater than expected. Also included in Fig. 6 is the case for which we assume that the MBL missing OH reactivity zero value is actually greater by the missing OH reactivity absolute uncertainty at 95% confidence (gray data). Comparing these two MBL cases shows that changes in the mean missing OH reactivity values affect only the median value and not the distribution skewness. Q-Q plots for ATom1 and ATom2 (not shown) are less dramatic, but still have the same characteristics: for measurements above 8 km, the high-value data are more normally distributed; for measurement in the MBL, ~20% of high-value data are greater than expected.

All of these statistical tests provide strong evidence for an abnormal amount of larger-than-expected missing OH reactivity in the MBL and up to 2-4 km altitude. It is possible that some individual outliers of the 1-minute data are due to anomalous OHR instrument issues. The few outlier data points at higher altitude could be due to these instrument issues but may also be due to pollution plumes that were encountered. However, it seems highly unlikely that ~20% of the higher-than-expected data at low altitudes could be caused by them. Thus, OH reactivity in the MBL is likely missing and needs to be further investigated.

### 3.3 Global Missing OH Reactivity in the Marine Boundary Layer

The frequent dips to below 200 m altitude gave 120 opportunities to examine the global distribution of missing OH reactivity. The measured OH reactivity averaged for each dip (Fig. 7(a, c, d) in the MBL (filled circles) is generally greater in the mid-latitudes and tropics than in the higher latitudes, reaching as high as 4-5 s$^{-1}$ over the Northern Hemisphere Pacific Ocean. More typical calculated values are $1.5\pm0.6$ s$^{-1}$, with relatively little variation. As a result, the missing OH reactivity values reflect the measured OH reactivity values.

Missing OH reactivity varied from ~0 s$^{-1}$ to ~2.5 s$^{-1}$ (Fig. 7 (b, d, f)). The lowest values occurred predominantly in the polar regions but also occasionally in the mid-latitudes and tropics. High values exceeding 1 s$^{-1}$ occurred predominantly over the Northern Hemisphere Pacific Ocean. The highest values occurred in ATom2, but values exceeding 2 s$^{-1}$ were also observed in ATom3. Missing OH reactivity appears to vary in both place and time.

A plot of missing OH reactivity as a function of latitude shows these variations in place and time (Fig. 8). There is a general tendency for missing OH reactivity to be greatest over the mid-latitudes and tropics and to decrease toward the poles. A sampling bias (Fig. 7) may be the reason for near-zero missing OH reactivity in the northern high latitudes and not in the southern high latitudes. However, the high missing OH reactivity over the Northern Hemisphere Pacific Ocean is exceptional.

A special note should be made regarding the northern Pacific data for ATom2. One flight (Anchorage, Alaska to Kailua-Kona, Hawai'i) accounts for missing OH reactivity values greater than ~2.5 s$^{-1}$. These points are anomalous in the context of all ATom OH reactivity measurements, and they do not correlate with the modeled influence from fires, convection, land, or the stratosphere. While present on all figures except Fig. 8, they were not included in the correlation analysis described below.

### 3.4 OH Reactivity Over Land

Of the approximately 120 dips in which OH reactivity measurements were made, 14% were over land (Figure 7). The majority of these were made in the Arctic, several over snow, ice, and tundra. As a result, the mean calculated OH reactivity was only 1.35 s$^{-1}$, while the mean measured OH reactivity was 1.4 s$^{-1}$ and the mean missing OH reactivity was -0.1 s$^{-1}$, which is essentially zero to well within uncertainties. Note, however, that there is little missing OH reactivity over most of the Arctic polar oceans as well as over the Arctic land, which means that missing OH reactivity is generally low over the entire colder Arctic region. The greatest measured missing OH reactivity was found on only one dip over the Azores, where the missing OH reactivity was ~2.5 s$^{-1}$ larger than the calculated OH reactivity.

### 3.5 Correlation of Missing OH Reactivity with Other Factors

From the procedure given in Section 2.6, missing OH reactivity has the four strongest correlations with butanal ($C_3H_7CHO$), dimethyl sulfide (DMS, $CH_3SCH_3$), formaldehyde (HCHO), and sea surface temperature (SST), as shown in Fig. 9. Missing OH reactivity also correlates with some modeled pptv-level butanal products, but at these low levels, these chemical species could not be the source of the missing OH reactivity. Interestingly, missing OH reactivity correlates only weakly with acetaldehyde and chlorophyll. These correlations suggest that the missing OH reactivity comes for an unknown VOC or OVOC that has HCHO and butanal as products and is co-emitted with DMS. The correlation with SST suggests an ocean source, as a higher temperature implies more emissions. Either biological activity of phytoplankton in the sea surface microlayer (Brooks and Thornton, 2017; Lana et al., 2011) or abiotic sea surface interfacial photochemistry (Brüggemann et al., 2018) could be the source of these VOCs and OVOCs.

### 3.6 Comparison to INTEX-B

HCHO is a good indicator for VOC photochemistry because it is an oxidation product for many VOCs. Thus, HCHO should correlate with missing OH reactivity. The ATom missing OH reactivity at the per-dip time resolution is compared to the Mao et al. (2009) missing OH reactivity below 2 km for times when NO is less than 100 pptv (Fig. 9). We use the per-dip time resolution of ~5 minutes in this comparison rather than per-flight to better align with the time resolution in Mao et al. (2009) of 3.5 minutes. The anomalously high missing OH reactivity from ATom2 are not included in the data for the ATom linear fit. The INTEX-B correlation coefficient between missing OH reactivity and HCHO ($R^2 = 0.58$) is better than the one found for ATom ($R^2 = 0.35$), but in the range of ATom HCHO (100 pptv – 500 pptv), the ATom correlation coefficient is larger.

The linear fit of the missing OH reactivity against HCHO data from Mao et al. (2009) is given as the solid red line in Fig. 9. If instead the pressure-dependent offset is used for Mao et al. (2009), then the resulting missing OH reactivity against HCHO follows the dashed red line. With the absolute INTEX-B offset uncertainty at ±0.5 s$^{-1}$ and the absolute ATom offset uncertainty at ±0.32 s$^{-1}$, both at the 68% confidence level, the linear fits for missing OH reactivity against HCHO in ATom and INTEX-B agree to within combined uncertainties. The ATom linear fit slope is only 2.7 standard deviations from the INTEX-B slope, but is 4.4 standard deviations from a line with zero slope, making it highly unlikely that missing OH reactivity is not correlated with HCHO. The INTEX-B and ATom slopes to the linear fits are not exactly the same. However, given the uncertainties, the HCHO dependence of the adjusted missing OH reactivity found in INTEX-B is consistent with that found for the ATom missing OH reactivity over the northern Pacific Ocean.

### 4 Discussion

Mao et al. (2009) calculated $HO_2/OH$ assuming that the cycling between OH and $HO_2$ was much greater than $HO_x$ production. That assumption is not valid for ATom because the low NO and OH reactivity values reduce the recycling to rates comparable to $HO_x$ production (Brune et al., 2020). On the other hand, by adding simple X photochemistry to the MCMv331 mechanism, as discussed in Section 2.5, it is possible to determine if the measured OH and $HO_2$ are consistent with observed missing OH reactivity. For case 1 in which there is no OH produced in the X oxidation sequence, the modeled

OH and HO$_2$ become 30-40% less than observed at altitudes below 2 km. On the other hand, if XO$_2$ and its products always autoxidize to produce OH (Crounse et al., 2013), then the modeled OH and HO$_2$ become 10-20% greater than observed. The optimum agreement between observed and modeled OH and HO$_2$ would require a partial recycling of OH, but without

knowing the identity of X, it is not possible to know the fraction of OH that should be recycled in the chemical mechanism. Thus, this analysis neither supports nor refutes the missing OH reactivity measurements.

Several recent studies provide evidence for an unknown VOC or OVOC emitted into the atmosphere from the ocean. Oceanic sources have also been proposed for butanes and pentanes in some regions (Pozzer et al., 2010) and for methanol

(Read et al., 2012). Measurements of biogenic VOCs in coastal waters found monoterpenes, C12-C15 n-alkanes, and several higher aldehydes that could contribute to enhanced OH reactivity (Tokarek et al., 2019).

Unexpectedly large abundances of acetaldehyde (CH$_3$CHO) have been observed in the marine boundary layer and the free troposphere (Singh et al., 2004; Millet et al., 2010; Read et al., 2012; Nicely et al., 2016; Wang et al., 2019) and the ocean is

the suspected source. While earlier measurements may have been compromised with interferences, recent measurements of unexpectedly large acetaldehyde abundances are supported by unexpectedly large abundances of peroxyacetic acid, which is produced almost exclusively through acetaldehyde oxidation (Wang et al., 2019). Wang et al. observed that the ocean effects on acetaldehyde were confined primarily to the MBL and were able to approximately model the vertical distribution by using direct ocean emissions of acetaldehyde. However, it is possible that some of the observed acetaldehyde was produced by

rapid oxidation of VOCs or OVOCs emitted from the ocean.

The missing OH reactivity is primarily in the MBL but often extends upward to as high as 2 km to 4 km in some dips. Above 4 km, the OH reactivity measurements are too near their LOD and thus too noisy to know if missing OH reactivity and acetaldehyde decrease the same way with altitude, but it is possible that they do. A similar decrease with altitude would

imply that the unknown reactant lives long enough to be distributed throughout the free troposphere. If, on the other hand, the missing OH reactivity is only in and just above the MBL, then the unknown reactant could have a much shorter lifetime. The lack of correlation between missing OH reactivity and acetaldehyde in the MBL suggests that the unknown reactant responsible for the missing OH reactivity is not necessarily connected only to an ocean source of acetaldehyde.

From Eq. (5) and the measured missing OH reactivity, the abundance of the chemical species X would typically be a few tenths of a ppbv, assuming that X is a sequiterpene with a typical reaction rate coefficient of $1 \times 10^{-10}$ cm$^3$ s$^{-1}$. The mean value for X is 0.26 $\pm$ 0.23 ppbv for the per-dip bins. If X is an alkane with a typical reaction rate coefficient of $2.3 \times 10^{-12}$ cm$^3$ s$^{-1}$, then its mixing ratio would need to be more than 10 ppbv.

If the unknown VOC is an alkane with a reaction rate coefficient with OH of $2.3 \times 10^{-12}$ cm$^3$ s$^{-1}$, then an unlikely large oceanic source of 340 Tg C yr$^{-1}$ would be necessary (Travis et al., 2020). Adding this much additional VOC reduces global modeled OH 20-50% along the ATom1 flight tracks, degrading the reasonable agreement with measured OH. Large sources of long-lived unknown VOCs, which do not have as large an impact on modeled OH, are also necessary to reduce but not resolve the discrepancies between measured and modeled acetaldehyde, especially in the Northern Hemisphere summer. These issues

between a global model and measured missing OH reactivity and acetaldehyde need to be resolved.

**5 Conclusion**

Measured OH reactivity significantly exceeds calculated OH reactivity in the marine boundary layer during ATom. This missing OH is most prominent over the northern and tropical Pacific Ocean where it had mean values of 0.4-0.7 s$^{-1}$ for the different ATom deployments, but rose to more than 2 s$^{-1}$ in some locations. These higher values correspond to ~0.26 ppbv of

a fast-reacting VOC, such as a sesquiterpene. The correlation of missing OH reactivity with formaldehyde, butanal, dimethyl sulfide, and sea surface temperature and the requirements for a smaller unknown reactive gas abundance and ocean source strength suggest that an ocean source of short-lived reactive gases, possibly VOCs or OVOCs, is responsible. This missing OH reactivity is qualitatively consistent with the observed unexpectedly large abundances of acetaldehyde, peroxyactic acid,

and other oxygenated VOCs. They may share the same cause. Finding this cause will require focused studies of detailed atmospheric composition in regions where missing OH reactivity and acetaldehyde excess are greatest.

## Data and Model Availability

The data and model used in this paper are publicly available:
- data: https://doi.org/10.3334/ORNLDAAC/1581
- model framework: https://github.com/airchem/F0AM
- MCMv331 chemical mechanism: http://mcm.leeds.ac.uk/MCM/

## Author Contribution

ABT, DOM, and WHB made the OH, $HO_2$, and OH reactivity measurements, performed the model runs, analyzed the data, and wrote the manuscript. GMW provided support of the use of the F0AM model framework used for the model runs. WHB, DOM, ABT, HMA, DRB, TPB, RC, JDC, BCD, GSD, JPD, JWE, SRH, TFH, RAH, EH, MJK, KM, FLM, JMN, JP, TBR, JMS, CS., APT, CT, KU, POW, GMW provided ATom measurements used for the modeling and reviewed and edited the manuscript.

## Competing Interests

The authors declare no financial or affiliation conflicts-of-interest.

## Funding

This study was supported by the NASA grant NNX15AG59A. This material is based upon work supported by the National Center for Atmospheric Research, which is a major facility sponsored by the National Science Foundation under Cooperative Agreement No. 1852977.

## Acknowledgements

The authors thank the NASA ATom management team, pilots, logistical support team, aircraft operations team, and fellow scientists. We thank the reviewers for their helpful comments on the initial submission.

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

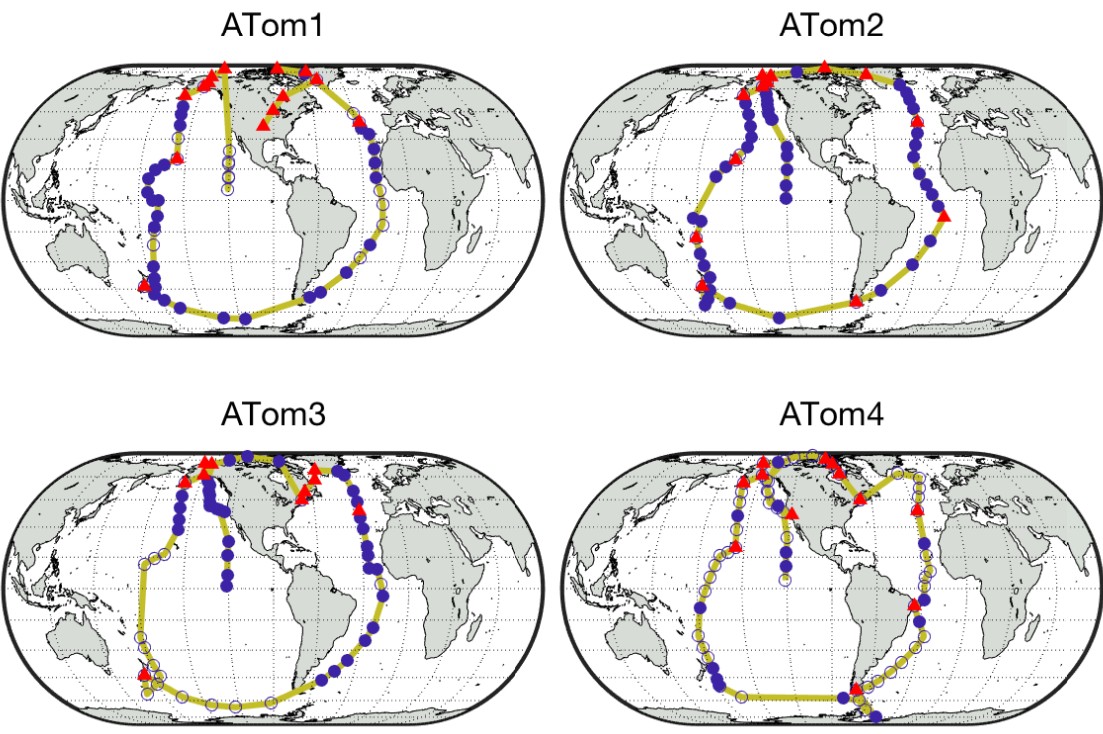


**Figure 1: Global ATom tracks (yellow lines) with indicators for the periods during which the DC-8 dipped into the boundary layer. Filled blue circles indicate points used for analysis; filled red triangles indicate dips when over land; unfilled blue circles indicate dips not used for analysis due to instrument calibrations or downtime.**

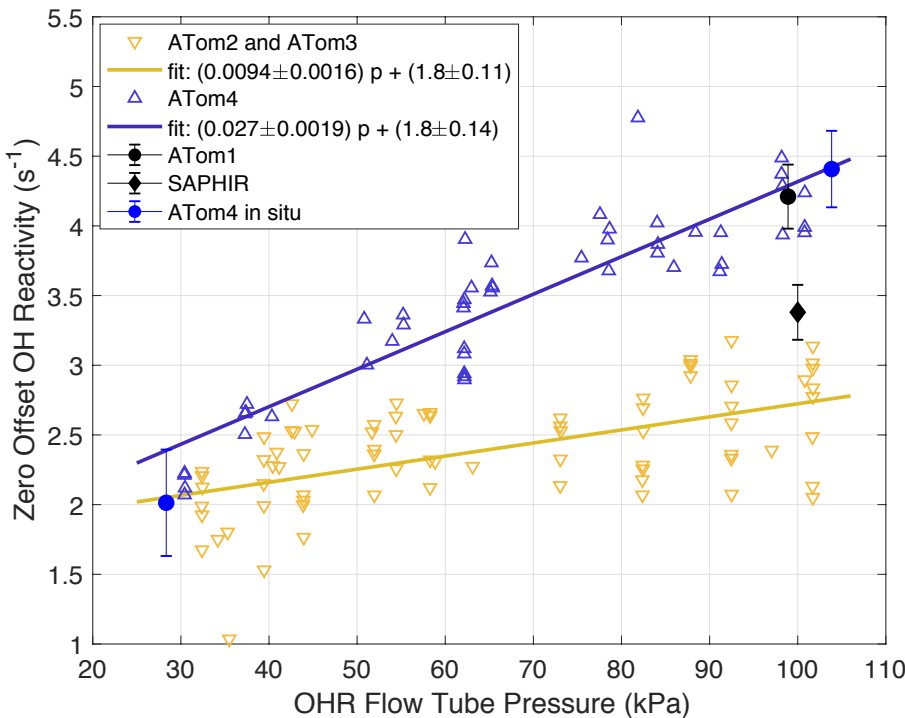

**Figure 2. Laboratory and in situ calibrations of OHR offset over 1-minute sums. The offset was calibrated only at ~100 kPa around ATom1 in 2015 and 2016 (black triangle). The offset was measured with a slightly different instrument configuration during the OH reactivity intercomparison study in 2015 (Fuchs et al., 2017). Offset calibrations performed in 2017 between ATom2 and ATom3 (yellow starts with linear fit (yellow line), in 2018 at the end of ATom4 (red circles) and linear fit (red line), and in flight are shown. Error bars are ±1 standard deviation of the mean. The ATom4 fit was used for ATom1 because the high-pressure laboratory calibrations were essentially the same.**


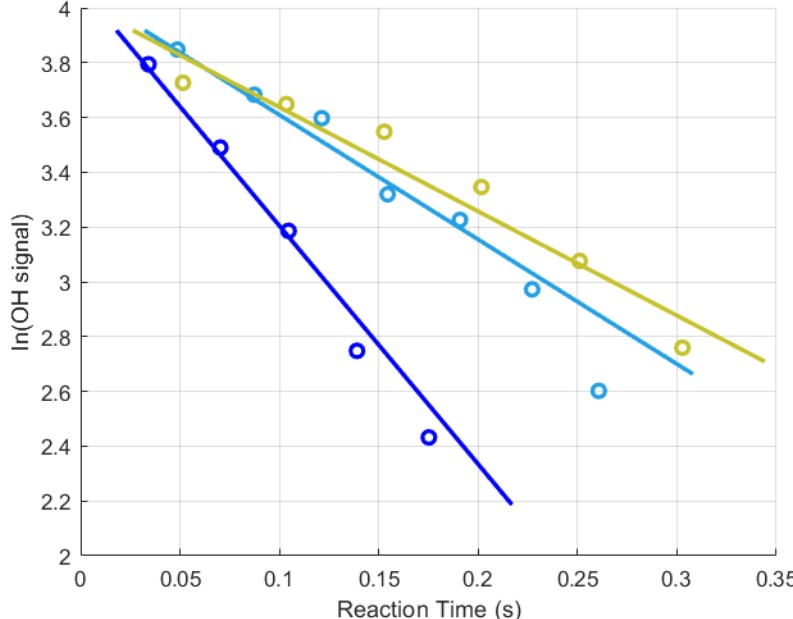

**Figure 3: Three in-flight decays for 1-minute sums of the OH signals. Decays were measured in the marine boundary layer and the individual 5 Hz data were binned by reaction times for clarity. When $k_{offset}$ is subtracted from the decays, their values become ~5 s$^{-1}$ (blue), ~3 s$^{-1}$ (teal), and ~2 (yellow) s$^{-1}$.**


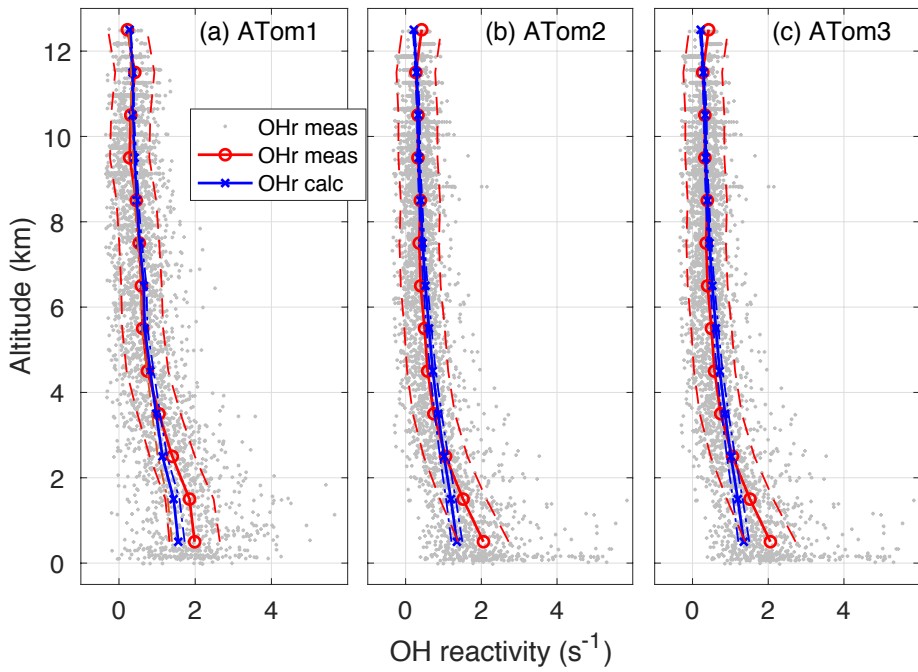

**Figure 4. OH reactivity versus altitude for ATom1 (August), ATom2 (February), and ATom3 (October). 1-minute measured OH**
**reactivity (grey dots), median measured OH reactivity (OHr meas) in 1-km altitude bins (red circle and line), and median calculated OH reactivity (OHr calc) in 1-km altitude bins (blue squares and line), and absolute OHR uncertainty (95% confidence level) for measured and calculated OH reactivity (dashed lines) are shown as a function of altitude.**

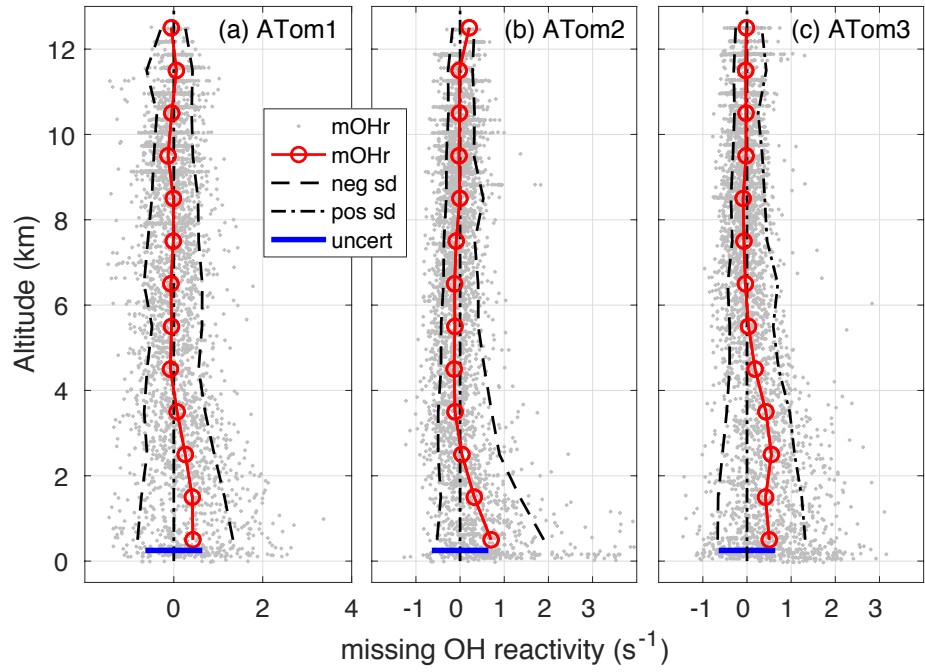


**Figure 5: Missing OH reactivity (mOHr) against altitude for ATom1 (August), ATom2 (February), and ATom3 (October). Grey dots are the OH reactivity calculated by subtracting calculated OH reactivity from measured OH reactivity. Median missing OH reactivity below 1 km altitude (red circles and lines) is comparable to the absolute uncertainty in the missing OH reactivity (blue bar, 95% confidence). The standard deviation of the negative missing OH reactivity data for each 1-km of altitude (left of the zero**
**line) and of the positive missing OH reactivity data (right of the zero line) are shown at the 95% confidence level and indicate the skewness in the missing OH reactivity data distribution below 2-4 km altitude.**

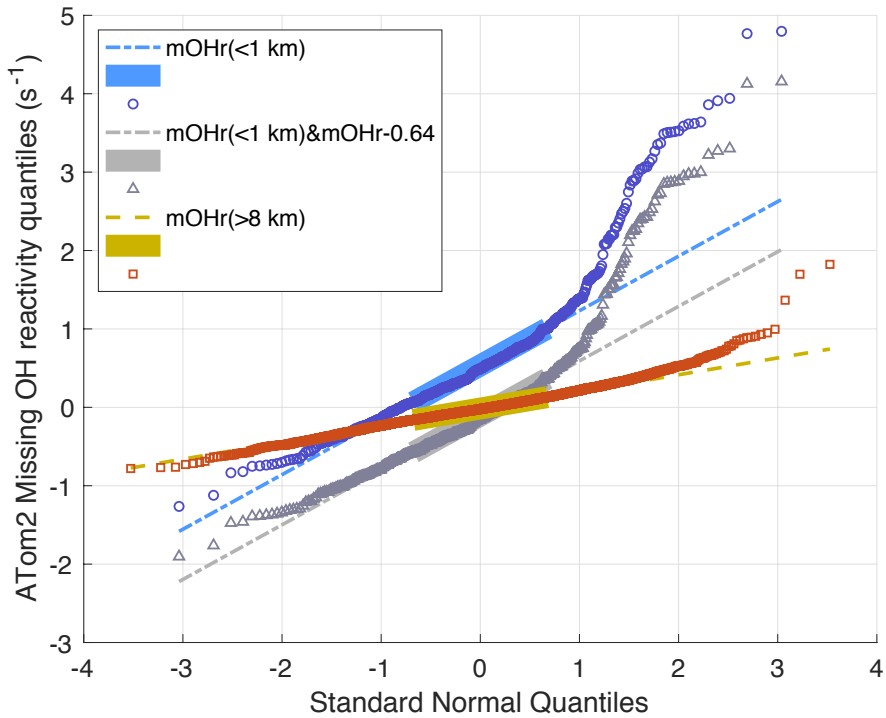


**Figure 6: ATom2 quantile-quantile plot for 1-minute missing OH reactivity values above 8 km (red squares) and below 1 km (blue circles) versus a normal distribution with a mean of 0 and a standard deviation of 1. The Q-Q plot for data taken below 1 km but with the median value shifted by 0.64 s⁻¹ (gray triangles) show the effect of an incorrect absolute missing OH reactivity median. The values lie along the dashed lines if the missing OH reactivity values are normally distributed. This Q-Q plot is for ATom2; the**

**Q-Q plots for ATom1 and ATom3 show less dramatic but similar behavior to that of ATom2.**

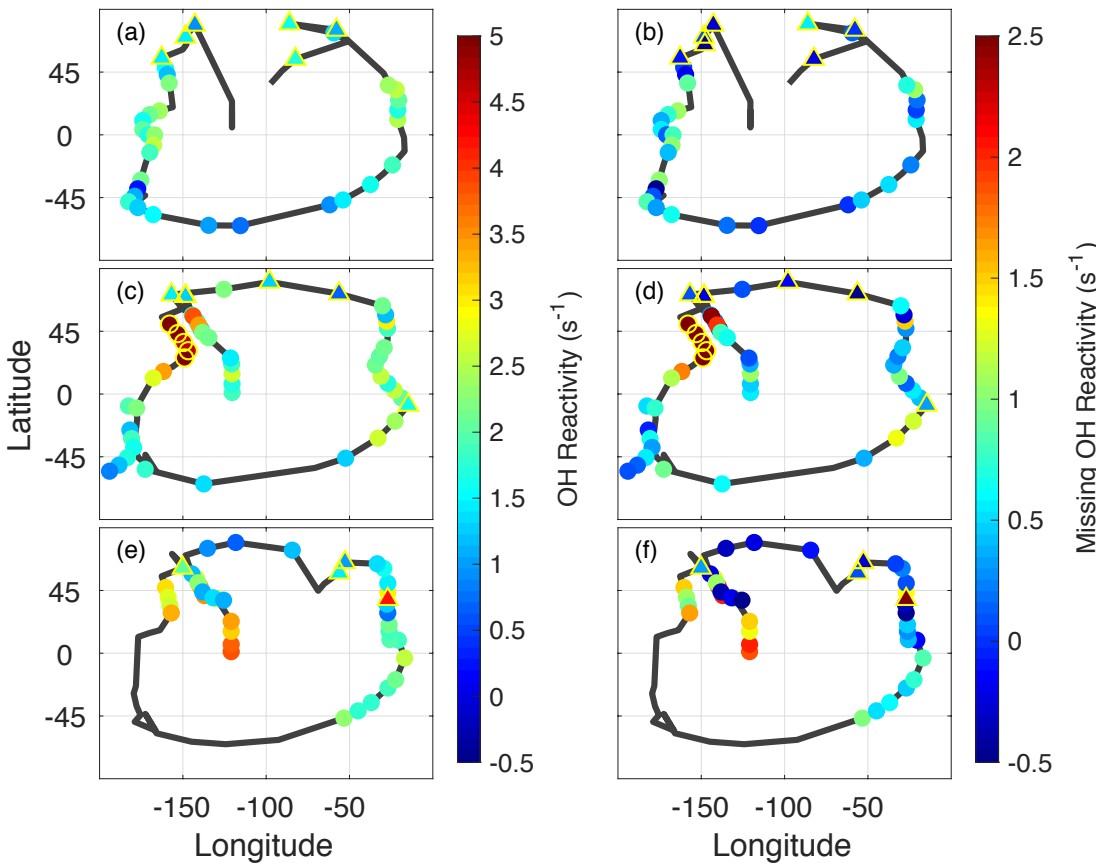

Figure 7. Global measured OH reactivity (a, c, and e) and missing OH reactivity (b, d, and f) for ATom1 (August), ATom2 (February), and ATom3 (October) at the per-dip time resolution. The black lines trace the flight path during each deployment, identical to the yellow tracks in Figure 1. Color indicates the measured OH reactivity (-0.5 to 5 s⁻¹ scale) and the missing OH reactivity (-0.5 to 2.5 s⁻¹ scale), while the yellow open circles indicate values in ATom2 above 2 s⁻¹ that were not included in the correlation analysis. Triangles outlined by yellow are overland values for both measured OH reactivity and missing OH reactivity.



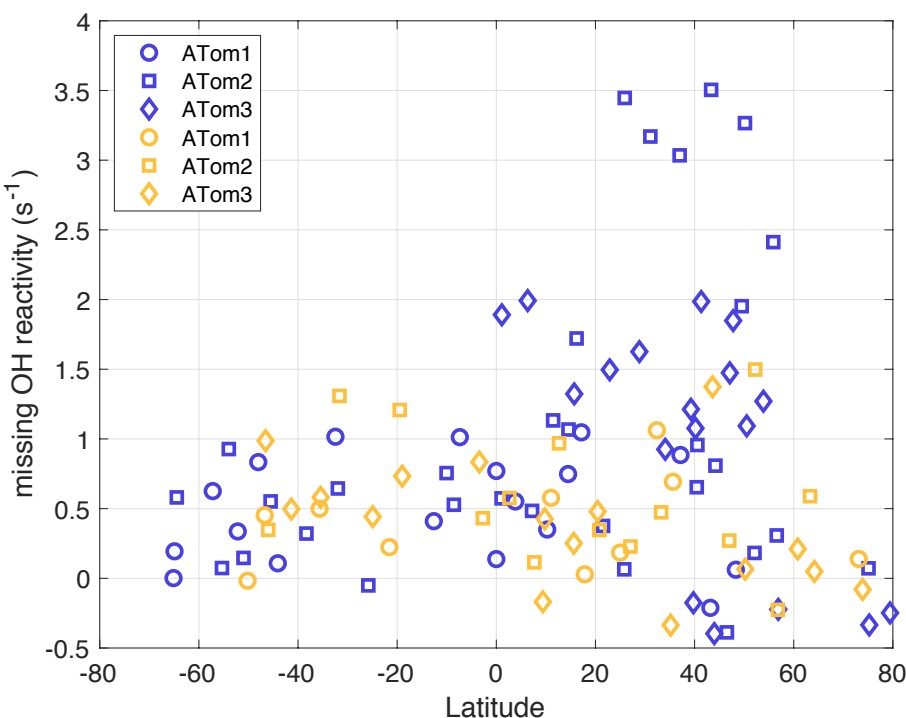

**Figure 8. Missing OH reactivity averaged per-dip versus latitude over the Pacific Ocean (blue) and the Atlantic Ocean (gold).**

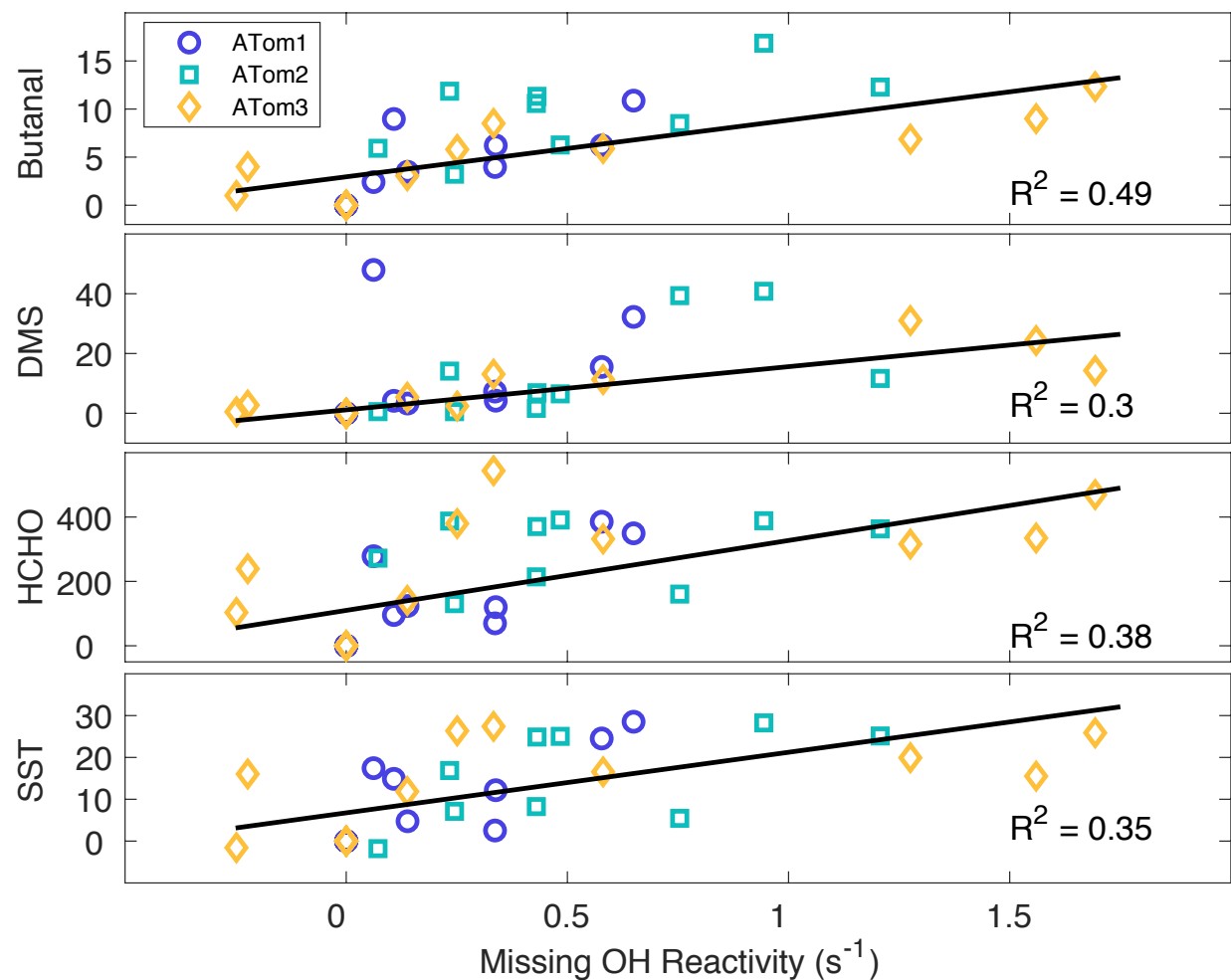

**Figure 8. The best correlations with missing OH reactivity for data at the per-flight resolution across all latitudes and hemispheres. The symbols are per-flight data for ATom1 (circles), ATom2 (squares), ATom3 (diamonds). Black lines are least squares fits to the per-flight data.**

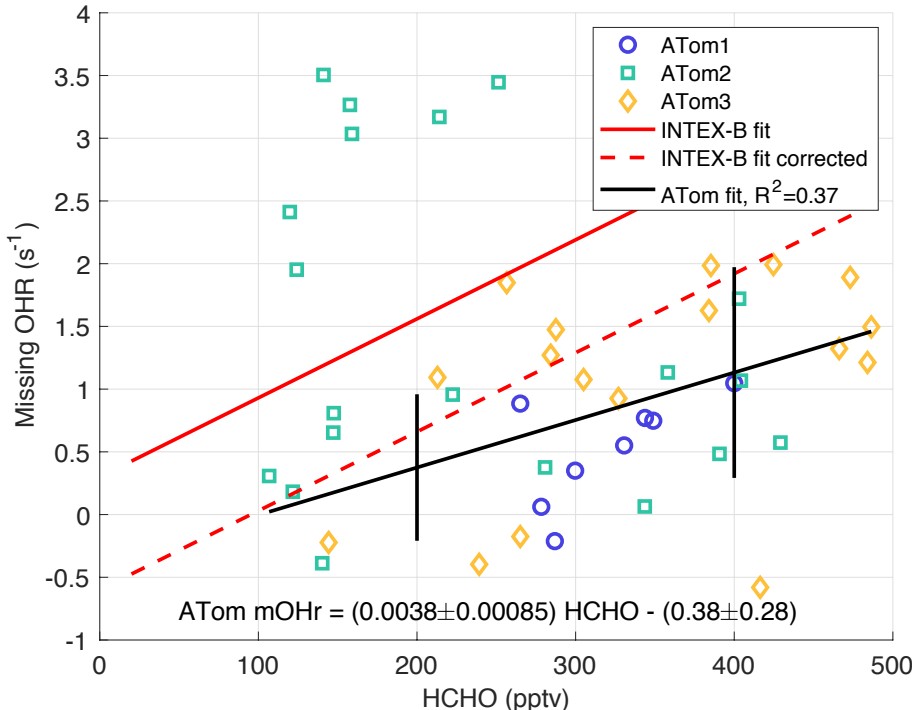

**Figure 9. Missing OH reactivity against HCHO for per-dip values in the MBL over the Northern Hemisphere Pacific Ocean. The ATom linear fit (black line) is shown with values for ATom1 (circles), ATom2 (squares), ATom3 (diamonds). The ATom linear fit is compared to the linear fit for missing OH reactivity values of Mao et. al (2009) (red line) and to this linear fit with an offset correction (red dashed line, see text). Uncertainty bars are the absolute uncertainty (95% confidence) of the missing OH reactivity. The statistical uncertainty in the slope and intercept of the linear fit are given in the equation on the figure.**

**Table 1: ATom campaign deployment seasons and start and end dates. Full details on stops can be found online (ATom, 2016).**

| Deployment | ATom1 | ATom2 | ATom3 | ATom4 |
|---|---|---|---|---|
| NH Season | Summer | Winter | Fall | Spring |
| Start Date | 28 July 2016 | 26 Jan 2017 | 28 Sept 2017 | 24 Apr 2018 |
| End Date | 22 August 2016 | 22 Feb 2017 | 26 Oct 2017 | 21 May 2018 |

780

**Table 2. Simultaneous measurements used to constrain the box model and calculate OH reactivity.**

| Measurement | Instrument | Uncertainty ($2\sigma$ confidence) | Reference |
|---|---|---|---|
| T | MMS | $\pm 0.5$ C | Chan et al., 1998 |
| p | | $\pm 0.3$ hPa | |
| $H_2O$ | DLH | $\pm 15\%$ | Diskin et al., 2003 |
| photolysis frequencies (30 measurements) | CAFS | $\pm (12\text{-}25)\%$, species dependent | Shetter and Mueller, 1999 |
| NO, $NO_2$ | NOyO3 | 6.6 pptv, 34 pptv | Ryerson et al., 2000 |
| $O_3$ | NOyO3[#] | 1.4 ppbv | Ryerson et al., 2000 |
| | UCATS | $\pm 1\% + 1.5$ ppbv | |
| CO | QCLS[#] | 3.5 ppbv | Santorini et al., 2014 |
| | NOAA Picarro | 3.6 ppbv | H. Chen et al., 2013 |
| | UCATS | 8.4 ppbv | |
| $H_2O_2$, $CH_3OOH$, $CH_3CO_3H$, $HNO_3$ | CIT CIMS | $\pm 30\% + 50$ pptv | Crounse et al., 2006 |
| SO2 | | $\pm 30\% + 100$ pptv | |
| HCOOH, BrO | NOAA CIMS | $\pm 15\% + 50$ pptv | Neuman et al., 2016 |
| $CH_4$ | NOAA Picarro[#] | 0.7 ppbv | Karion et al., 2013 |
| | UCATS | 23.6 ppbv | |
| | PANTHER | 34.6 ppbv | |
| HCHO | NASA ISAF | $\pm 10\% \pm 10$ pptv | Cazorla et al., 2015 |
| methyl nitrate, ethyl nitrate, isoprene, acetylene, ethylene, ethane, propane, i-butane, n-butane, i-pentane, n-pentane, n-hexane, n-heptane, benzene, toluene, methyl chloride, methylene chloride, chloroform, methyl bromide, methyl chloroform, perchloroethene, 1,2-dichloroethane, DMS | UCI WAS | $\pm 10\%$ | Colman et al., 2001 |
| methanol, formaldehyde, acetaldehyde, ethyl benzene, toluene, methacrolein, methyl ethyl ketone, methyl tert-butyl ether, ethanol, acetone, 2-methylpentane, 3-methylpentane, 2,2,4-trimethylpentane, isobutene+1-butene, m-xylene+p-xylene, o-xylene, tricyclene, limonene+D3-carene, propanal, butanal, acrolein | TOGA | $\pm 15\text{-}50\%$ (acetaldehyde: $\pm 20\%$) | Apel et al., 2015 |

# Primary measurement. Other measurements fill gaps in primary measurement.

**Table 3: Simple X photochemistry added to the photochemical mechanism to test for effects of X on modeled OH and HO$_2$**

| Reaction | Reaction rate coefficient $(cm^3\ s^{-1})$ |
|---|---|
| case 1: $X + OH \rightarrow XO_2$ | $1\times10^{-10}$ |
| case 2: $X + OH \rightarrow XO_2 + OH$ | |
| $\quad X + O_3 \rightarrow XO_2$ | $1\times10^{-16}$ |
| $\quad XO_2 + NO \rightarrow HO_2 + NO_2 + prod$ | $3\times10^{-12}\exp(300/T)$ |
| $\quad XO_2 + HO_2 \rightarrow XOOH$ | $8.6\times10^{-13}\exp(700/T)$ |
| $\quad XOOH + h\upsilon \rightarrow XO + OH$ | $J_{CH3OOH}\ (s^{-1})$ |
| $\quad XOOH + OH \rightarrow XO_2$ | $2.9\times10^{-12}\exp(-160/T)$ |

790