# Peer review of "Missing OH Reactivity in the Global Marine Boundary Layer"

_Atmospheric Chemistry and Physics, 2019_

## Referee Comment (RC1) · Anonymous Referee #1 · 13 Nov 2019

This paper presents OH reactivity measurements from the ATom aircraft project, providing a substantial dataset in the under-studied marine boundary layer which will no doubt help to improve our understanding of the global oxidation capacity. A comparison of measured OH reactivity with modelled OH reactivity in this region seems to demonstrate that there is missing OH reactivity and the authors attribute this to an ocean source of short-lived reactive gases. As well as a number of minor comments, I have a few queries on the analyses performed to demonstrate that the missing OH reactivity in the MBL is statistically significant. Once these questions have been addressed, I am suggesting this manuscript is published in ACP.

Pg 1, line 36: Define OHR

Pg 1, line 38: Calculated or modelled OH reactivity?

The amount of 'missing' reactivity often depends on the completeness of the individual OH sinks that were measured alongside. Although not the primary focus of this paper, it would be informative to know if the OH reactivity budget could be closed in the boundary layer over land?

Pg 3, paragraph 3: Given the sparsity of MBL OH reactivity observations, I suggest the authors expand their discussion (in section 4 on the earlier Mao et al study) to include the Pfannersill et al study which reports higher MBL OH reactivities and higher missing OH reactivities than observed during ATom.

Table 2: Was NO2 measured during the project? If it was, but was not used to constrain the model, could the authors provide a comment on the level of agreement between measured and modelled NO2?

Pg 5, line 140: 'background signal' I presume the authors mean the 'OH offline' signal? As it reads, however, this 'background signal' may be confused with kbackground.

Pg 5, line 141: Did the ratio of the flow of carrier gas to the flow of ambient air vary with altitude? If it did, the authors should comment on the impact impurities in the carrier gas may have at high and low altitudes respectively. Could a change in the flow ratios explain the observed pressure dependence presented in Fig 2?

Pg 5, line 148: what NO concentration do the authors class as 'high NO'?

Pg 5, line 156: Do the authors expect the low molecular weight VOCs present in the PAM chamber to form particles?

Pg 5, line 159: What do the authors mean by 'media'

Pg 6, line 198: Was a pressure dependent background applied to all the OH reactivity data?

Pg 7, line 209: '..only 0.2 s-1' vs Pg 6, line 187: '0.25 – 0.3 s-1'

Pg 7, section 2.3: How are photolysis rates treated in the model?

Pg 9, line 279: 'Some extreme outlier points were removed..' the authors should comment on the approach they chose to remove data – was this data flagged as potentially having a problem?

Pg 9, line 287 and figure 4: Some of the OH reactivity data measured above 10 km do not match the model calculated OH reactivity exactly. Why?

Pg 9, line 292: Do the authors expect ambient HO2 to make it into the flow tube without being lost on inlet lines? Given the fast rate coefficient employed in the model for the reaction of CH3O2+OH, did CH3O2 not contribute a significant sink for OH?

Pg 10, line 309: 4 km is much higher than the MBL and for ATom 3 there seems to be statistically significant missing OH reactivity up to 6 km (fig 5). Was there any evidence of long-range transport of pollution in these regions that could contribute to OH reactivity (and missing OH reactivity) during these flights?

Pg 10, line 321: The authors should also comment of the 8 – 12 km data in figure 6. Some of these points also lie above the red dashed line. The number of 8 – 12 km points lying above the line is less than in the 0 – 4 km data, but is this simply because the 10 – 12 km data was set to match the modelled reactivity? I suggest that this analysis is conducted on 8 – 10 km data only and also 0 – 2 km and 2 – 4 km separately.

Pg 11, fig 7: I don't think the trend in missing reactivity with latitude is best illustrated by figure 7. Do the authors see a reasonable correlation if they plot missing reactivity vs latitude in a scatter plot?

Pg 11, line 339: The authors should make it clear which figure these data have been included in.

Pg 12, line 367: What were the typical NO concentrations during the campaigns? Table 3: I presume that the reactions from line 3 onwards apply to both Case 1 and Case 2? As the Table is set out, however, it currently looks like only 1 reaction (X + OH = XO2)

is added for Case 1. If 'XO2 + NO = HO2' is included as a reaction, shouldn't 'HO2 + NO = OH' also be considered?

Pg 13, line 392: I think the authors mean 'missing' reactivity here.

Pg 13, 405 onwards: To determine the source strength, both the lifetime and the ambient concentration of the two species needs to be considered. So, although the calculated concentration of the alkane is ∼43 times greater than the calculated concentration of the sesquiterpene, the lifetime of the alkane is ∼43 longer, so the source strength to maintain the calculated concentrations of both species should be the same.

Pg 13, line 407: the Travis et al., 2019 is missing from the reference list.

Pg 13, line 412: 's-1' - superscript '-1'

Pg 13, line 413: '∼0.5 ppb' or '0.26 ppb'?
* * *

---

## Referee Comment (RC2) · Anonymous Referee #2 · 26 Nov 2019

==========================

Summary

==========================

Thames and co-authors present OH reactivity measurements over the remote oceans from the ATom campaigns. They use a 0D box model constrained to other ATom observations to interpret the OHR data, specifically focusing on 'missing' OHR in the marine boundary layer.

The dataset is the first of its kind, spanning the globe, and is collected under analytically challenging conditions (very clean air). The team is to be commended for the work that went into collecting this dataset. It represents a great contribution to the community

that I expect will be used by many future researchers.

My main comments on the manuscript have to do with its treatment of uncertainties and statistics. Since we are looking at very low OHR conditions, the missing OHR values are also low and pushing the uncertainty limits. I feel that the paper needs a more sophisticated and robust treatment of uncertainty statistics (including in the modeled OHR, which is itself constrained to measurements) in order to provide a convincing case that the missing OHR values are indeed robust. Once that is done the paper should be published in ACP.

==========================

General comments

==========================

Along with uncertainties in the OHR measurement itself, the "modeled" OHR also has uncertainties. It is constrained to trace gas measurements, which have their own uncertainties. It predicts unmeasured species using rate coefficients that have uncertainties. It seems to me that in order to judge whether the missing OHR is statistically robust, these uncertainties need to be fully propagated through the modeled OHR derivation. Then one could do a proper statistical comparison of the measured and modeled OHR values.

164: "these variations were tracked with measurements of the OHR instrument background in the laboratory". I am confused by this because an earlier statement (line 141) appears to indicate that the background was measured every measurement cycle. Please clarify.

180-184: how can we be confident that the pressure dependence of the calibration is prescribed accurately enough to define (for example) a mean 0.5 1/s discrepancy? The relatively large amount of scatter in Figure 2 (e.g., for the ATom-1/4 calibration) by itself does not by itself inspire confidence in this respect. Given the small OHR discrepancies that are discussed later, I feel the paper needs a more rigorous discussion of the background and calibration uncertainties, along with a quantitative analysis of how these propagate onto the end results, for us to have confidence in the findings.

It doesn't appear that a pressure-dependent calibration curve was performed at the time of ATom-1. How are we confident that the 2018 curve fits the ATom-1 data given the 2-year separation in time?

190-194: A pressure-invariant offset is being applied to the measurements based on model output for the upper free troposphere. Please indicate the magnitude of this offset that is being applied (e.g., compared to the inferred missing OHR magnitudes that are discussed later). Is there reason to believe that this offset is in fact pressure-independent as assumed?

207-208: "Therefore, in each ATom phase, the total uncertainty in the OH reactivity is dominated by the instrument background uncertainty." My interpretation of this is that we should be considering the errors as primarily systematic rather than random. I.e., the campaign-specific background at any given pressure is a single constant quantity that we can define to 1-sigma of 0.4 1/s. And therefore that uncertainty is not reduced by temporal averaging of the campaign measurements: the background uncertainty is the same (0.4 1/s) whether we are considering 1 measurement or thousands. Is my interpretation correct? If so then I don't believe 1-sigma is an appropriate metric, since sigma is a measure of variability rather than of certainty about the central value. A more appropriate metric would be the 95% confidence interval about the pressure-dependent backgrounds – for example, obtained via boostrap analysis of the data in Figure 2.

299: "The median measured OH reactivity equals the median model-calculated OH re-activity to within $\pm 1\sigma$ statistical uncertainty", see above, sigma is a measure of scatter rather than uncertainty

Figure 2, "Darker grey points indicate OH reactivity values greater than the 1-sigma

uncertainty in the MBL." Wording is unclear here. At first I thought it meant "missing OH reactivity values greater than the 1-sigma...", but from the plot it looks like the colored values are just those where the actual OH reactivity is 1-sigma above the median value. Please clarify.

305-311 and Figure 5: There is some conflation of spread and uncertainty here. First, the "For missing OH reactivity to be meaningful, some missing OH reactivity points must be much greater than the statistical spread of the OH reactivity measurements." A bit oddly worded, rather one should say that to be meaningful, the missing OH reactivity should exceed the statistical uncertainty of the OH reactivity measurements. Spread and uncertainty are not the same thing. Similarly, in the Figure 5 caption: "Dotted black lines represent +/-2-sigma uncertainty derived from a median of the missing OH reactivity values greater than 4 km." If the lines are just twice the SD they are showing the spread, not the uncertainty. And wording-wise it is not clear what "2-sigma uncertainty derived from a median" means. Finally, "About 95% of all points above 4 km are within that phase's $2\sigma$ uncertainty bands, which is consistent with a statistically normal distribution." – again mixing up variability with uncertainty.

313-327 and Figure 6: This (qq-plots and t-test) is a nice demonstration that the missing OHR data above and below 4km follow differing statistical distributions. Please discuss the robustness of this finding in view of i) the statistical uncertainty of the pressure-dependent background corrections in Figure 2, ii) the propagated uncertainty in the modeled OHR, and iii) the assumption of a pressure-invariant offset (line 190-194). Second, the figure is only showing ATom-2 data but the text (by not mentioning this) implies that all of the data from ATom-1, 2, and 3 have this feature. Is that the case?

333-334: "The latitudinal dependence implies that air or sea temperature or other latitude-dependent factors contribute to missing OH reactivity." Also, the highest missing OHR values fall in the NH, implying that the generally higher abundance of trace gases in the NH plays a role ... right?

342: "the main correlations that stand out are..." please be more precise in your language here, are these the 4 variables with the highest correlations?

342-350 and Figure 8: please discuss whether these correlations persist when the campaigns are considered individually.

363: "the 1-sigma confidence level", please see earlier comments about confidence intervals. What is needed here is a statement of whether the slopes agree to within (say) 95% confidence based on a bootstrap / monte carlo test. In the same way, please also indicate whether the slope is significantly different than zero.

372-373: "become substantially less than observed" and "become greater than observed", please be quantitative

My understanding is that the NO2 measurements during ATom have high uncertainty. Is that right? Are you using the measured NO2 or is this being predicted by the model from other species?

==========================

Minor / technical / language corrections

==========================

There are some minor grammatical errors throughout; please do a careful proof-reading.

29: "which IS 0.5 s-1 larger"

35-36: "for much of the free troposphere", awkward, suggest "throughout much ..."

45: suggest "with THE hydroxyl RADICAL"

46-47: "is lost by the sum of the reaction frequencies", wording is not quite right b/c the loss is via the chemical reactions themselves, the frequencies just determine how fast that occurs. Suggest "is lost at a rate determined by the sum..."

47: suggest "This sum of loss frequencies is called. . ."

68: "exceeded the calculated AMOUNT by"

69: VOC not defined

71: "in A northern Michigan forest"

75: "20%, which is approximately the uncertainty". But doesn't this percentage depend on the absolute OHR amount?

103: as stated later this 0.4 1/s LOD is for 1-minute averages, consider specifying that here

149: "in high NO environments", please specify the approximate NO level at which this effect becomes relevant

190: "1-minute sums", perhaps this should be "1-minute averages"

Figure 1: I don't know that it is helpful to include ATom-4 in this Figure given that the data is not ultimately used in the analyses that follow.

230: "and other measurements were used to fill gaps in the primary measurement". Can you please add a few words to be more specific here? E.g., "linear regression to other measurements"?

250: Need to specify assumed OH level giving this 1-hour lifetime

Figure 2: If I understand Figure 2 correctly, the blue fit is being used for both ATom-1 and ATom-4, is that correct? If so, the legend should be relabeled to make this more clear.

Fig 2: "The median OH reactivity of 500m altitude bins is shown formeasured OH reactivity (blue line, with $1\sigma$ error bars)", in fact the error bars are only shown at 2km increments, suggest clarifying in caption

270: "These legs. . ." awkward wording

270-271: suggest stating range of MBL heights during ATom.

290: some representative OHR ranges would be helpful here.

318: "The missing OH reactivity values measured below 4 km altitude lie along the red dashed line" I think you mean "above 4 km" here.

Figures 1, 4, 5, 7: I recognize that this information is also in Table 1, but it would be helpful to your reader if you indicated the time-frame of each ATom deployment somewhere on these figures.

Table 1: a single season is given for each ATom deployment, but ATom covered both hemispheres.

336-340: do you suspect instrumental factors here?

340: "While present on some figures", please be specific

Figure 8, "at the per-flight time resolution" is unclear, I think you mean that each point is an average over all the data for a given flight?

356-357: wording is awkward here

---

## Author Comment (AC1) · 30 Jan 2020

Response to the referees' comments

We thank the referees for useful comments. Their comments are in italics, followed by our responses in Arial font and our changes to the manuscript in Times-Roman font.

In addition to responding to the referees' comments, we have also expanded the discussion of global missing OH reactivity in the MBL, improved some of the wording and consistency, and corrected typos.

Anonymous Referee #1.

*This paper presents OH reactivity measurements from the ATom aircraft project, providing a substantial dataset in the under-studied marine boundary layer which will no doubt help to improve our understanding of the global oxidation capacity. A comparison of measured OH reactivity with modelled OH reactivity in this region seems to demonstrate that there is missing OH reactivity and the authors attribute this to an ocean source of short-lived reactive gases. As well as a number of minor comments, I have a few queries on the analyses performed to demonstrate that the missing OH reactivity in the MBL is statistically significant. Once these questions have been addressed, I am suggesting this manuscript is published in ACP.*

*Pg 1, line 36: Define OHR*
It is now defined as "OH Reactivity" in the abstract and again in the first paragraph of section 2.2.

*Pg 1, line 38: Calculated or modelled OH reactivity?*
We removed the words "value of" to say "The mean measured OH reactivity …".

*The amount of 'missing' reactivity often depends on the completeness of the individual OH sinks that were measured alongside. Although not the primary focus of this paper, it would be informative to know if the OH reactivity budget could be closed in the boundary layer over land?*

Thanks to the referee for this suggestion. We have added a Section 3.3 OH Reactivity Over Land and have included the measured and missing OH reactivity values per dip in a new version of Figure 7. The new text is the following:

"Of the approximately 120 dips in which OH reactivity measurements were made, 14% were over land (Figure 7). The majority of these were made in the Arctic, several over snow, ice, and tundra. As a result, the median calculated OH reactivity was only 1.35 s$^{-1}$, while the median measured OH reactivity was 1.4 s$^{-1}$ and the median missing OH reactivity was -0.1 s$^{-1}$, which is essentially zero to well within uncertainties. Note, however, that there is little missing OH reactivity over most of the Arctic polar oceans as well as over the Arctic land, which means that missing OH reactivity is generally low over the entire colder Arctic region. The greatest measured missing OH reactivity was found on only one dip over the Azores, where the missing OH reactivity was ~2.5 s$^{-1}$ larger than the calculated OH reactivity."

Unfortunately, these measurements do not provide contribute to the evidence supporting the hypothesis that the missing OH reactivity over the oceans is due to ocean gaseous emissions because they were primarily in the Arctic where there was little missing OH reactivity.

*Pg 3, paragraph 3: Given the sparsity of MBL OH reactivity observations, I suggest the authors expand their discussion (in section 4 on the earlier Mao et al study) to include the Pfannersill et al study which reports higher MBL OH reactivities and higher missing OH reactivities than observed during ATom.*

Pg 3, paragraph 2 was already devoted to discussing the Pfannersill et al. study. We have enhanced it by adding more detail:

"One regime that has yet to be adequately investigated is the remote marine boundary layer (MBL) and the free troposphere above it, which comprises 70% of the global lower troposphere. Two prior studies measured OH reactivity in the MBL. The most recent was shipborne across the Mediterranean Sea, through the Suez Canal, and into the Arabian Gulf in summer 2017 (Pfannerstill et al., 2019). Several portions of this journey were heavily influenced by petrochemical activity or ship traffic, while others were relatively clean. Median measured OH reactivity for the different waterways ranged from 6 $s^{-1}$ to 13 $s^{-1}$, while median calculated OH reactivity ranged from 2 $s^{-1}$ to 9 $s^{-1}$. When more than 100 measured chemical species were included in the calculated OH reactivity, the difference between the measured and calculated OH reactivity was reduced to being with measurement and calculation uncertainty for some regions, but significant missing OH reactivity remained for other regions. In the cleaner portions of the Mediterranean and Adriatic Seas, the calculated OH reactivity of ~2 $s^{-1}$ was below the instrument's limit of detection (LOD = 5.4 $s^{-1}$)."

We note that essentially all ATom OH reactivity measurements in the MBL were far below the LOD of the instrument used in the Pfannerstill et al. research.

*Table 2: Was NO2 measured during the project? If it was, but was not used to constrain the model, could the authors provide a comment on the level of agreement between measured and modelled NO2?*

$NO_2$ was measured and is now included in Table 2. Measured $NO_2$ did not always agree with modeled $NO_2$ by as much as 30-50%. However, with a few exceptions, $NO_2$ was less than 40 pptv and accounted for less than 0.5% of the total calculated OH reactivity. Therefore, any issue with $NO_2$ has a negligible effect on the calculated OH reactivity.

*Pg 5, line 140: 'background signal' I presume the authors mean the 'OH offline' signal? As it reads, however, this 'background signal' may be confused with kbackground.*

We agree with the referee and have changed the sentence to read: "…while the OH detection system switches the laser wavelength to off resonance with OH to measure the signal background."

*Pg 5, line 141: Did the ratio of the flow of carrier gas to the flow of ambient air vary with*

*altitude? If it did, the authors should comment on the impact impurities in the carrier gas may have at high and low altitudes respectively. Could a change in the flow ratios explain the observed pressure dependence presented in Fig 2?*

We thank the referee for this question. The ratio of the wand flow, which is constant, to the total reaction tube flow does change with pressure, resulting in an increase in the hypothesized contaminant concentration with increasing pressure (i.e., decreasing altitude). In fact, the differences in the two fitted curves in Figure 2 can be mainly explained by this pressure dependence. We have added a paragraph after paragraph 6 in Section 2.2 that says:

"The difference in the linear fit to the offset calibration for ATom1 and ATom 4 and the linear fit to the offset calibration for ATom2 and ATom3 is pressure dependent (Fig. 2). The standard volume airflow in the wand was constant, but the ambient volume flow in the flow tube decreased by a factor of ~2 as the flow tube pressure increased from 30 kPa to 100 kPa. As a result, the contamination concentration from the wand air also increased a factor of ~2 as flow tube pressure increased. This pressure-dependent contamination concentration explains much of the difference between the two fitted lines and provides evidence that contamination in the wand flow was a substantial contributor to the changes in the zero offset between ATom1/ATom4 and ATom2/ATom3. The good agreement between the fit for ATom2/ATom3 and the offset calibrations of Mao et al. (2009), who used ultra-high purity $N_2$, suggests that the zero air for ATom2 and ATom3 had negligible contamination."

*Pg 5, line 148: what NO concentration do the authors class as 'high NO'?*

We change this statement to "…in environments where NO is greater than a few ppbv, …"

Pg 5, line 156: Do the authors expect the low molecular weight VOCs present in the PAM chamber to form particles?

No, we do not. But our experience is that when there are low-molecular weight VOCs in contaminated ambient air, there are also higher molecular-weight VOCs as well. We have also added a sentence describing another test that we neglected to mention, which was to do some runs with high-purity $N_2$ as a comparison. The results were the same, but the PAM chamber test proved to be the more sensitive of the two. We add a sentence:
"The results of this test were consistent with those obtained by substituting the air from the zero air generator with high purity nitrogen."

Pg 5, line 159: What do the authors mean by 'media'
We add a parenthetical statement: "(Perma Pure ZA-Catalyst – Palladium on Aluminum Oxide)".

*Pg 6, line 198: Was a pressure dependent background applied to all the OH reactivity data?*

Yes.

*Pg 7, line 209: '..only 0.2 s-1' vs Pg 6, line 187: '0.25 – 0.3 s-1'*

One is the calculated OH reactivity using the flow tube pressure and temperature and the other is the calculated OH reactivity using ambient pressure and temperature. We now make this difference clear by changing the one on line 209 to "The OH reactivity from the model at the ambient temperature and pressure rarely exceeded 2 s$^{-1}$ in the planetary boundary and was only 0.2 s$^{-1}$ in the free troposphere."

*Pg 7, section 2.3: How are photolysis rates treated in the model?*

The measured photolysis rates were used in the model to calculate the unmeasured but modeled chemical species. To get the calculated OH reactivity at the flowtube temperature and pressure, the model was initialized by constraining the measured and modeled chemical species for each time step and then running the model at the flow tube temperature and pressure for 1 second at a fixed OH value in order to generate the rate coefficients and concentrations needed to calculate the OH reactivity that would have been see in the flow tube conditions.

*Pg 9, line 279: 'Some extreme outlier points were removed..' the authors should comment on the approach they chose to remove data – was this data flagged as potentially having a problem?*

We no longer remove any points before doing the correlations. This difference in approach changes the correlation values somewhat but did not change the variables for which the correlations are most significant.

*Pg 9, line 287 and figure 4: Some of the OH reactivity data measured above 10 km do not match the model calculated OH reactivity exactly. Why?*

We made the mean value of the measured OH reactivity and the mean value of the calculated OH reactivity the same over the altitude range from 10-12 km, but that does not mean that they will be the same at every altitude between 10 km and 12 km.

*Pg 9, line 292: Do the authors expect ambient HO2 to make it into the flow tube without being lost on inlet lines? Given the fast rate coefficient employed in the model for the reaction of CH3O2+OH, did CH3O2 not contribute a significant sink for OH?*

The contributions to the calculated OH reactivity by $HO_2$+OH and $CH_3O_2$+OH are comparable and small. In the lowest 2 km, they contribute less than 3% to the mean total calculated OH reactivity. Assuming that $HO_2$ and $CH_3O_2$ are lost in the OHR instrument and its sampling lines, we plotted the missing OH reactivity with their contributions subtracted. The differences in the plots was barely perceptible and none of the numbers for statistical significance or correlations changed. We do not know for certain that these radicals are lost in the instrument or sampling lines, however we do know that sticky chemical species such as HCHO, HOOH, and $CH_3OOH$ have no

measurable loss. Thus, we choose to retain the figures without this correction. In Section 3 Results, we add the sentence:

"It is possible that $HO_2$ and $CH_3O_2$ are lost in the Teflon sampling lines or the OHR instrument before they can be measured, but their mean contribution to the calculated OH reactivity is less than 3% and can be ignored in our analysis."

*Pg 10, line 309: 4 km is much higher than the MBL and for ATom 3 there seems to be statistically significant missing OH reactivity up to 6 km (fig 5). Was there any evidence of long-range transport of pollution in these regions that could contribute to OH reactivity (and missing OH reactivity) during these flights?*

Pollution plumes were encountered over a range of altitudes during the missions. It is possible that these few points are due to missing OH reactivity in those encounters, but it is also possible are just statistical outliners due to aircraft maneuvers, short-lived instrument issues, or clouds. We will look at these when we do an analysis of individual flights.

*Pg 10, line 321: The authors should also comment of the 8 – 12 km data in figure 6. Some of these points also lie above the red dashed line. The number of 8 – 12 km points lying above the line is less than in the 0 – 4 km data, but is this simply because the 10 – 12 km data was set to match the modelled reactivity? I suggest that this analysis is conducted on 8 – 10 km data only and also 0 – 2 km and 2 – 4 km separately.*

Matching the means of the measured and calculated OH reactivity for data abouve 10 km has nothing to do with the deviations from the normal distribution in the Q-Q plot. Thus using 10-12 km is appropriate, although we have changed this range to being above 8 km. The Q-Q plots for 8-10 km and 10-12 km are very similar. So we will use 8-12 km for the counterexample on the distribution. We also focus on 0-1 km, which is closest to the height of the MBL but still contains enough data.

*Pg 11, fig 7: I don't think the trend in missing reactivity with latitude is best illustrated by figure 7. Do the authors see a reasonable correlation if they plot missing reactivity vs latitude in a scatter plot?*

We have now added a plot of per-dip missing OH reactivity as a function of latitude (Figure 8) in addition to the global view in Figure 7. We think that both have value.

*Pg 11, line 339: The authors should make it clear which figure these data have been included in.*

Done

*Pg 12, line 367: What were the typical NO concentrations during the campaigns? Table 3: I presume that the reactions from line 3 onwards apply to both Case 1 and Case 2? As the Table is set out, however, it currently looks like only 1 reaction (X + OH = XO2)*

*is added for Case 1. If 'XO2 + NO = HO2' is included as a reaction, shouldn't 'HO2 + NO = OH' also be considered?*

HO$_2$+NO is in MCMv331, so it does not need to be added. NO in the lowest 2 km was 7 $\pm$7 pptv, very low. If X +OH is not important, then nothing in the subsequent chemistry is going to be important because the production of X is not going to be important. You can think of X+OH as representing that reaction and all the subsequent reaction as far as OH reactivity goes.

*Pg 13, line 392: I think the authors mean 'missing' reactivity here.*

Fixed.

*Pg 13, 405 onwards: To determine the source strength, both the lifetime and the ambient*
*concentration of the two species needs to be considered. So, although the calculated concentration of the alkane is 43 times greater than the calculated concentration of the sesquiterpene, the lifetime of the alkane is 43 longer, so the source strength to maintain the calculated concentrations of both species should be the same.*

The referee is correct.

We modify the paragraph:
"If the unknown VOC is an alkane with a reaction rate coefficient with OH of $2.3 \times 10^{-12}$ cm$^3$ s$^{-1}$, then an unlikely large oceanic source of 340 Tg C yr$^{-1}$ would be necessary (Travis et al., 2020). Adding this much additional VOC reduces global modeled OH 20-50% along the flight tracks, degrading the reasonable agreement with measured OH. Large sources of long-lived unknown VOCs, which do not have as large an impact on modeled OH, are also necessary to reduce but not resolve the discrepancies between measured and modeled acetaldehyde, especially in the Northern Hemisphere summer. These issues between a global model and measured missing OH reactivity and acetaldehyde need to be resolved."

*Pg 13, line 407: the Travis et al., 2019 is missing from the reference list.*

Travis et al., 2020, was recently submitted to ACPD. It is now listed in the references.

*Pg 13, line 412: 's-1' - superscript '-1'*

Fixed

*Pg 13, line 413: '0.5 ppb' or '0.26 ppb'?*

Fixed

Anonymous Referee #2

*Thames and co-authors present OH reactivity measurements over the remote oceans from the ATom campaigns. They use a 0D box model constrained to other ATom observations to interpret the OHR data, specifically focusing on 'missing' OHR in the marine boundary layer. The dataset is the first of its kind, spanning the globe, and is collected under analytically challenging conditions (very clean air). The team is to be commended for the work that went into collecting this dataset. It represents a great contribution to the community that I expect will be used by many future researchers. My main comments on the manuscript have to do with its treatment of uncertainties and statistics. Since we are looking at very low OHR conditions, the missing OHR values are also low and pushing the uncertainty limits. I feel that the paper needs a more sophisticated and robust treatment of uncertainty statistics (including in the modeled OHR, which is itself constrained to measurements) in order to provide a convincing case that the missing OHR values are indeed robust. Once that is done the paper should be published in ACP.*
=============================
*General comments*
=============================
*Along with uncertainties in the OHR measurement itself, the "modeled" OHR also has uncertainties. It is constrained to trace gas measurements, which have their own uncertainties. It predicts unmeasured species using rate coefficients that have uncertainties. It seems to me that in order to judge whether the missing OHR is statistically robust, these uncertainties need to be fully propagated through the modeled OHR derivation. Then one could do a proper statistical comparison of the measured and modeled OHR values.*

We agree and have done a much more thorough analysis of all the uncertainties. We answer this question about the model uncertainty and the measurement uncertainty together as an answer to the line 180-184 comments.

*164: "these variations were tracked with measurements of the OHR instrument background in the laboratory". I am confused by this because an earlier statement (line 141) appears to indicate that the background was measured every measurement cycle. Please clarify.*

The confusion results from our using the word 'background" for two different measurements. We now state (old line 141): "the OH detection system switches the laser wavelength to off resonance with OH to measure the signal background." to make it clear that its background is in the laser signal. We have renamed the OH reactivity instrument background as the "offset" throughout the manuscript for consistency.

The first sentence in Section 2.3 now reads:
"The OHR offset varied between the 4 ATom phases due to changes in the zero-air generator performance and between research flights due to internal contamination from pre-flight conditions. These changes were tracked with measurements of the OHR instrument offset in the laboratory and, for ATom4, in situ during several flights."

*180-184: how can we be confident that the pressure dependence of the calibration is prescribed accurately enough to define (for example) a mean 0.5 1/s discrepancy? The relatively large amount of scatter in Figure 2 (e.g., for the ATom-1/4 calibration) by itself does not by itself inspire confidence in this respect. Given the small OHR discrepancies that are discussed later, I feel the paper needs a more rigorous discussion of the background and calibration uncertainties, along with a quantitative analysis of how these propagate onto the end results, for us to have confidence in the findings.*
*It doesn't appear that a pressure-dependent calibration curve was performed at the time of ATom-1. How are we confident that the 2018 curve fits the ATom-1 data given the 2-year separation in time?*

We have extensively re-analyzed our OHR offset data and better quantified our estimate the total uncertainty in the missing OH reactivity measurements. We created a new Section 2.3 OH reactivity measurement offset calibrations and another new Section 2.4 Missing OH reactivity uncertainty analysis.

We need to stress that doing the in-flight offset calibration was extremely difficult, involving crawling into the forward cargo bay of an aircraft bouncing 500 ft above the ocean surface, all the while trying to adjust a regulator to keep the air flow in the OHR flowtube constant for three minutes as the cylinder pressure slowly decreases. We add the following to Section 2.3:

"The difficulty of maintaining steady calibration conditions in flight during ATom4 caused the large in situ calibration error. The standard deviation of these offset calibrations is 0.75 s$^{-1}$, which is 2.5 to 3 times larger than the SD obtained for ambient measurements in clean air for the same altitude and number of measurements, indicating that the atmospheric measurement precision is much better than could be achieved in these difficult offset calibrations. Yet even with this lower precision, the median offset at high and low pressure agree with the linear fit of the laboratory calibrations to within 20% at low pressures and 3% at high pressure."

We are confident that the ATom1 pressure dependence is the same as that for ATom4 because of the excellent agreement at 100 kPa and the knowledge that the low pressure offset measurements in the laboratory, in situ for ATom4, and Mao et al. (2009) are all at or slightly above 2 s$^{-1}$. This similiarity between ATom1 and ATom4 suggests that the contamination that plagued ATom4 also plagued ATom1.

We add the following in Section 2.3:

"For ATom1, the offset was calibrated at only 97 kPa prior to the mission, but it is in excellent agreement with the offset calibrated for ATom4. We can safely assume that the ATom4 offset slope can be applied to ATom 1 because all offset calibrations performed at low OHR flowtube pressures, even those of Mao et al. (2009), are ~2 s$^{-1}$."

*190-194: A pressure-invariant offset is being applied to the measurements based on model output for the upper free troposphere. Please indicate the magnitude of this offset that is being applied (e.g., compared to the inferred missing OHR magnitudes*

*that are discussed later). Is there reason to believe that this offset is in fact pressure independent as assumed?*

This paragraph has been rewritten with the following explanation:

"The OHR offset varied slightly from flight to flight because the variable air quality produced by the zero-air generator. This flight-to-flight variation was tracked and the OH reactivity offset was corrected by the following procedure. The OH reactivity calculated from the model at the OHR instrument's temperature and pressure (sec Sect 2.3) was 0.25-0.30 s[-1] for the upper troposphere during all ATom phases and latitudes. The offset calibrations were adjusted in the range of $0.34 \pm 0.32$ s[-1] for each research flight by a pressure-invariant offset that was necessary to equate the median measured and model-calculated OH reactivity values for data taken above 8 km altitude. If this offset correction is not used for all altitudes, then the OH reactivity in the 2- 8 km range varies unreasonably from flight-to-flight, even going significantly negative at times. In effect, we used the upper troposphere as a clean standard in order to fine-tune $k_{offset}$, just as Mao et al. (2009) did."

*207-208: "Therefore, in each ATom phase, the total uncertainty in the OH reactivity is dominated by the instrument background uncertainty." My interpretation of this is that we should be considering the errors as primarily systematic rather than random. I.e., the campaign-specific background at any given pressure is a single constant quantity that we can define to 1-sigma of 0.4 1/s. And therefore that uncertainty is not reduced by temporal averaging of the campaign measurements: the background uncertainty is the same (0.4 1/s) whether we are considering 1 measurement or thousands. Is my interpretation correct? If so then I don't believe 1-sigma is an appropriate metric, since sigma is a measure of variability rather than of certainty about the central value. A more appropriate metric would be the 95% confidence interval about the pressure dependent backgrounds – for example, obtained via boostrap analysis of the data in Figure 2.*

We have completely revised this analysis and description. We agree that systematic (i.e., absolute) errors affect the missing OH reactivity values and use our knowledge of uncertainties in the several components going into the missing OH reactivity calculation to perform a sensitivity (i.e. error propagation) analysis. This description has become the first paragraph in the new Section 2.4:

"The uncertainty for missing OH reactivity in the MBL at the 68% confidence level comes from four components: the decay measurement itself; the offset as determined by the slope and intercepts of the fits to the laboratory OH reactivity offset calibrations (Fig. 2); the flight-to-flight offset variation as judged by fitting the measured OH reactivity to the model-calculated OH reactivity at 8-12 km altitude; and the model calculations. First, the uncertainty in decay fit is approximately $\pm 7.5\%$, which for a typical OH reactivity measurement in the MBL of ~2 s[-1], would give an uncertainty of $\pm 0.15$ s[-1]. Second, the uncertainty in the OH reactivity offset in the MBL is found from the sum of the slope uncertainty times the OHR flow tube pressure, which is ~100 kPa in the MBL, ($\pm 0.16$ s[-1]) and the intercept uncertainty ($\pm 0.11$ s[-1]). The two uncertainties are assumed to be correlated. Third, the uncertainty in the flight-to-flight offset variation is the standard deviation of the mean for each high altitude short level leg ($\pm 0.15$ s[-1]). Fourth, the uncertainty of the model-calculated OH reactivity was determined by Eq. 4:

$$\Delta k_{OH}^{calc} \ (s^{-1}) = \sqrt{\sum [(k_i \Delta x_i)^2 + (\Delta k_i x_i)^2]} \qquad (4)$$

where $k_i$ are the reaction rate coefficients and $x_i$ are the OH reactant concentrations. The rate coefficient uncertainties come from Burkholder et al. (2016) and the chemical species uncertainties come from Table 2 and Brune et al. (2020). For the 11 chemical species responsible for 95% of the total OH reactivity in the MBL, this uncertainty is $\pm 0.08$ s$^{-1}$. The square root of the sum of the squares of all these uncertainties yields a total uncertainty for the MBL missing OH reactivity of $\pm 0.32$ s$^{-1}$ at the 68% confidence level."

Note that this more careful calculation of the mOHR absolute uncertainty is somewhat less than our earlier estimate of $\pm 0.4$ s$^{-1}$.

*299: "The median measured OH reactivity equals the median model-calculated OH reactivity*
*to within 1 statistical uncertainty", see above, sigma is a measure of scatter*
*rather than uncertainty*

The confidence level is directly related to the probability of occurrence, which is measured by the standard deviation in a normal distribution, if the uncertainty is approximately normally distributed. None-the-less we have changed our notation on the confidence levels to percentages.

*Figure 2, "Darker grey points indicate OH reactivity values greater than the 1-sigma uncertainty in the MBL." Wording is unclear here. At first I thought it meant "missing OH reactivity values greater than the 1-sigma: : :", but from the plot it looks like the colored values are just those where the actual OH reactivity is 1-sigma above the median value. Please clarify.*

Removed.

*305-311 and Figure 5: There is some conflation of spread and uncertainty here. First, the "For missing OH reactivity to be meaningful, some missing OH reactivity points must be much greater than the statistical spread of the OH reactivity measurements." A bit oddly worded, rather one should say that to be meaningful, the missing OH reactivity should exceed the statistical uncertainty of the OH reactivity measurements. Spread and uncertainty are not the same thing. Similarly, in the Figure 5 caption: "Dotted black lines represent +/-2-sigma uncertainty derived from a median of the missing OH reactivity values greater than 4 km." If the lines are just twice the SD they are showing the spread, not the uncertainty. And wording-wise it is not clear what "2-sigma uncertainty derived from a median" means. Finally, "About 95% of all points above 4 km are within that phase's 2 uncertainty bands, which is consistent with a statistically normal distribution." – again mixing up variability with uncertainty.*

We have redone the analysis and made it much more rigorous and statistically sound. This section has been extensively rewritten:

**"3.2 Missing OH Reactivity: Statistical Evidence**

A better approach is to find the missing OH reactivity for each measurement time point and then look at the mean values. The missing OH reactivity is plotted as a function of altitude for ATom1, ATom2, and ATom3 (Fig. 5). The mean missing OH reactivity is set to 0 s$^{-1}$ for 8-12 km, but remains

near to 0 down to 2-4 km, where it then increases. The 1-minute measurements are a good indicator of the measurement precision, which is $\pm 0.35$ s$^{-1}$ for ATom1 and $\pm 0.25$ s$^{-1}$ for ATom2 and ATom3.

[revised manuscript text omitted]

*333-334: "The latitudinal dependence implies that air or sea temperature or other latitude-dependent factors contribute to missing OH reactivity." Also, the highest missing OHR values fall in the NH, implying that the generally higher abundance of trace gases in the NH plays a role : : : right?*

Removed.

*342: "the main correlations that stand out are: : :" please be more precise in your language here, are these the 4 variables with the highest correlations?*

This sentence has been rewritten:
"From the procedure given in Section 2.6, missing OH reactivity has the four strongest correlations with …".

*342-350 and Figure 8: please discuss whether these correlations persist when the campaigns are considered individually.*

Please see the discussion in Section 2.6. For the different ATom phases, the correlations could be different with the much sparser individual data sets. The criteria used required correlations for each ATom phase using both the pre-dip and pre-flight data sets. By requiring correlations simultaneously for multiple methods, we are confident that we have found robust correlations.

To see what the relationship is for different ATom phases, please look at the revised Figure 8. In general, you can see the correlation for each phase by focusing only on its points.

*363: "the 1-sigma confidence level", please see earlier comments about confidence intervals. What is needed here is a statement of whether the slopes agree to within (say) 95% confidence based on a bootstrap / monte carlo test. In the same way, please also indicate whether the slope is significantly different than zero.*

To find the absolute uncertainty, we chose to differentiate Eq. 3 and use propagation of error analysis to get our absolute uncertainty estimates, which is just as good as using the standard deviation of a distribution obtained by Monte Carlo simulations.

We do not have the data for Mao et al. (2009) and so cannot find the uncertainty in the slope and intercept to their fit. However, we add the uncertainties at the 95% confidence level ($2\sigma$ on a normal probability distribution) for the absolute missing OH reactivity uncertainty to show that it is possible that the two observations are the same.

We did calculate the standard deviation of the slope for the ATom fit and have included it in the figure. We have also revised the paragraph:

"The linear fit of the missing OH reactivity against HCHO data from Mao et al. (2009) is given as the solid red line in Fig. 9. If instead the pressure-dependent offset is used for Mao et al. (2009), then the resulting missing OH reactivity against HCHO follows the dashed red line. With the absolute INTEX-B offset uncertainty at $\pm0.5$ s$^{-1}$ and the absolute ATom offset uncertainty at $\pm0.32$ s$^{-1}$, both at the 68% confidence level, the linear fits for missing OH reactivity against HCHO in ATom and INTEX-B agree to within combined uncertainties. The ATom linear fit slope is only 2.7 standard deviations from the INTEX-B slope, but is 4.4 standard deviations from a line with zero slope, making it highly unlikely that missing OH reactivity is not correlated with HCHO. The INTEX-B and ATom slopes to the linear fits are not exactly the same. However, given the uncertainties, the HCHO dependence of the adjusted missing OH reactivity found in INTEX-B is consistent with that found for the ATom missing OH reactivity over the northern Pacific Ocean."

372-373: "become substantially less than observed" and "become greater than observed", please be quantitative

We have revised the sentence:

"For case 1 in which there is no OH produced in the X oxidation sequence, the modeled OH and HO$_2$ become 30-40% less than observed at altitudes below 2 km. On the other hand, if XO$_2$ and its products autoxidize to produce OH (Crounse et al., 2013), then the modeled OH and HO$_2$ become 10-20% greater than observed."

My understanding is that the NO2 measurements during ATom have high uncertainty. Is that right? Are you using the measured NO2 or is this being predicted by the model from other species?

NO$_2$ was measured and is now included in Table 2. Measured NO$_2$ did not always agree with modeled NO$_2$ by as much as 30-50%. However, with a few exceptions, NO$_2$ was less than 40 pptv and accounted for less than 0.5% of the total calculated OH reactivity. Therefore, any issue with NO$_2$ has a negligible effect on the calculated OH reactivity.

==========================
*Minor / technical / language corrections*
==========================
*There are some minor grammatical errors throughout; please do a careful proofreading.*
*29: "which IS 0.5 s-1 larger"*
Fixed.

*35-36: "for much of the free troposphere", awkward, suggest "throughout much : : :"*
We think this statement is fine as it is and are leaving it unchanged,

*45: suggest "with THE hydroxyl RADICAL"*
Changed.

*46-47: "is lost by the sum of the reaction frequencies", wording is not quite right b/c the loss is via the chemical reactions themselves, the frequencies just determine how fast that occurs. Suggest "is lost at a rate determined by the sum: : :"*
Corrected.

*47: suggest "This sum of loss frequencies is called: : :"*
Corrected.

*68: "exceeded the calculated AMOUNT by"*
"measured OH reactivity" is an amount. We will leave this sentence as is.

*69: VOC not defined*
Fixed

*71: "in A northern Michigan forest"*
Fixed

*75: "20%, which is approximately the uncertainty". But doesn't this percentage depend on the absolute OHR amount?*
Yes, but the goal is to close the budget, no matter what the absolute amount is. So the percentage is the most important quantity, not the absolute value for accomplishing this goal.

*103: as stated later this 0.4 1/s LOD is for 1-minute averages, consider specifying that here*
The new sentence is more specific:
"Although the calculated OH reactivity in the middle-to-upper troposphere is less than the OH reactivity instrument's LOD, which is $\sim\pm 0.4$ s$^{-1}$ for 1-minute averages when both absolute uncertainty and measurement variability are taken into account, this instrument can measure OH reactivity in and just above the MBL."

*149: "in high NO environments", please specify the approximate NO level at which this effect becomes relevant*
Done. It's approaching 10 ppbv.

*190: "1-minute sums", perhaps this should be "1-minute averages"*
Removed.

*Figure 1: I don't know that it is helpful to include ATom-4 in this Figure given that the data is not ultimately used in the analyses that follow.*

We want to include it even if we do not use it in the analysis.

*230: "and other measurements were used to fill gaps in the primary measurement".
Can you please add a few words to be more specific here? E.g., "linear regression to
other measurements"?*
We basically just substituted one measurement for the other. It must be noted that with
a few exceptions, the measurements that we had to substitute were in excellent
agreement. We chose the ones with the highest resolution, but substituted in ones with
slightly lower when the higher resolution measurements were not available. This
substitution accounted for less than 10% of the total time.

*250: Need to specify assumed OH level giving this 1-hour lifetime*
We have modified this sentence to read:
"…which gives X a lifetime of about an hour for the typical daytime [OH] of $\sim 3 \times 10^6$ cm$^{-3}$."

*Figure 2: If I understand Figure 2 correctly, the blue fit is being used for both ATom-1
and ATom-4, is that correct? If so, the legend should be relabeled to make this more
clear.*
The new figure caption now reads:
"Figure 2. Laboratory and in situ calibrations of OHR offset over 1-minute sums. The offset was calibrated
only at ~100 kPa around ATom1 in 2015 and 2016 (black triangle). The offset was measured with a slightly
different instrument configuration during the OH reactivity intercomparison study in 2015 (Fuchs et al.,
2017). Offset calibrations performed in 2017 between ATom2 and ATom3 (yellow starts with linear fit
(yellow line), in 2018 at the end of ATom4 (red circles) and linear fit (red line), and in flight (blue dots with
error bars indicating the range of 75% of the data) are shown. The ATom4 fit was used for ATom1 because
the high-pressure laboratory calibrations were essentially the same."

*Fig 2: "The median OH reactivity of 500m altitude bins is shown formeasured OH
reactivity (blue line, with 1 error bars)", in fact the error bars are only shown at 2km
increments, suggest clarifying in caption*
I believe that Referee #2 is referring to Figure 4. This figure and its caption have been
redone.

*270: "These legs: : :" awkward wording*
We add the word "level" to be consistent with the wording in the preceding sentence.

*270-271: suggest stating range of MBL heights during ATom.*
We have corrected these sentences to read:
"Each per-dip bin is a single value representing an average of the missing OH reactivity as the DC-8 flew
a level leg at 160 m. These level legs were generally well in the MBL because its height was greater than
160 m 85% of the time."

*290: some representative OHR ranges would be helpful here.*
We have added the non-restrictive qualifier: "which is typically 10-50 s$^{-1}$." after the
reference.

*318: "The missing OH reactivity values measured below 4 km altitude lie along the red*

*dashed line" I think you mean "above 4 km" here.*
This comment is no longer relevant because the plot and its description have been completely changed.

*Figures 1, 4, 5, 7: I recognize that this information is also in Table 1, but it would be helpful to your reader if you indicated the time-frame of each ATom deployment somewhere on these figures.*
We added the month in parentheses to the captions for Figs. 4, 5, and 7.

*Table 1: a single season is given for each ATom deployment, but ATom covered both hemispheres.*
We added "NH" to "Season" to make it clear.

*336-340: do you suspect instrumental factors here?*
No. We checked the raw decays very carefully and they were good.

340: "While present on some figures", please be specific
The sentence now reads:
"While present on all figures except Fig. 8, they were not included in the correlation analysis."

Figure 8, "at the per-flight time resolution" is unclear, I think you mean that each point is an average over all the data for a given flight?
The caption now reads"
"Figure 8. The best correlations with missing OH reactivity for data at the per-flight resolution across all latitudes and hemispheres. The symbols are per-flight data for ATom1 (circles), ATom2 (squares), ATom3 (diamonds). Black lines are least squares fits to the per-flight data."

*356-357: wording is awkward here*
The sentence has been reworded to read:
"The INTEX-B correlation coefficient between missing OH reactivity and HCHO ($R^2 = 0.58$) is better than the one found for ATom ($R^2 = 0.35$), but in the range of ATom HCHO (100 pptv – 500 pptv), the ATom correlation coefficient is larger."